# GENERATE-AND-FILTER: CAUSAL REGENERATION OF DATASETS FOR CROSS-DOMAIN RECOMMENDATION

## ABSTRACT

Cross-domain recommendation (CDR) has emerged as an effective strategy to mitigate data sparsity and cold-start challenges by transferring knowledge from a source domain to a target domain. Despite recent progress, two key issues remain: **(i) Sparse overlap.** In real-world datasets such as Amazon, the proportion of users active in both domains is extremely low, significantly limiting the effectiveness of many state-of-the-art CDR approaches. **(ii) Negative transfer.** Existing methods primarily address this problem at the model level, often assuming that logged interactions are unbiased and noise-free. In practice, however, recommender data contain numerous spurious correlations, and this issue is exacerbated in CDR due to domain heterogeneity. To address these challenges, we propose a dataset regeneration framework. First, we leverage a prediction model to generate a pool of high-confidence candidate interactions to link non-overlapping target-domain users and source-domain items. Second, inspired by causal inference, we introduce a filtering process designed to prune spurious interactions. This process identifies and removes not only noisy edges created during generation but also those from the original dataset, retaining only the interactions that have a positive causal effect on the target-domain performance. Through these two processes, we can regenerate a source-domain dataset that exhibits a tighter coupling and a more explicit causal connection with the target domain. By integrating our method with three representative recommendation backbones—LightGCN, BiTGCF, and CUT—we show that it significantly boosts their predictive accuracy on the target domain, achieving substantial gains of up to 23.81% in Recall@10 and 22.22% in NDCG@10.

## 1 INTRODUCTION

Data sparsity and cold start problems have been critical challenges in recommender systems, as user interactions with items are often limited, especially for new users or items Lika et al. (2014); Kang et al. (2019). Traditional single-domain recommendation models struggle to provide accurate recommendations when faced with sparse user-item interactions because their operations are limited in isolated data silos, preventing them from leveraging rich user signals that may exist in other, more data-abundant domains. A natural and promising idea to overcome this is to break down these barriers and transfer knowledge across different domains.

Following this idea, Cross-Domain Recommendation (CDR) emerges as a promising direction to tackle this challenge by transferring knowledge from one (source) domain to improve recommendation accuracy in another (target) domain. Through CDR, the aim is to leverage richer user interaction signals in the source domain, thus alleviating data sparsity and cold start issues in the target domain Singh & Gordon (2008a); Gao et al. (2013); Hu et al. (2018); Liu et al. (2020); Yang et al. (2024). The core challenge of CDR lies in how to effectively transfer knowledge from the source domain to the target domain. Some early attempts at CDR are

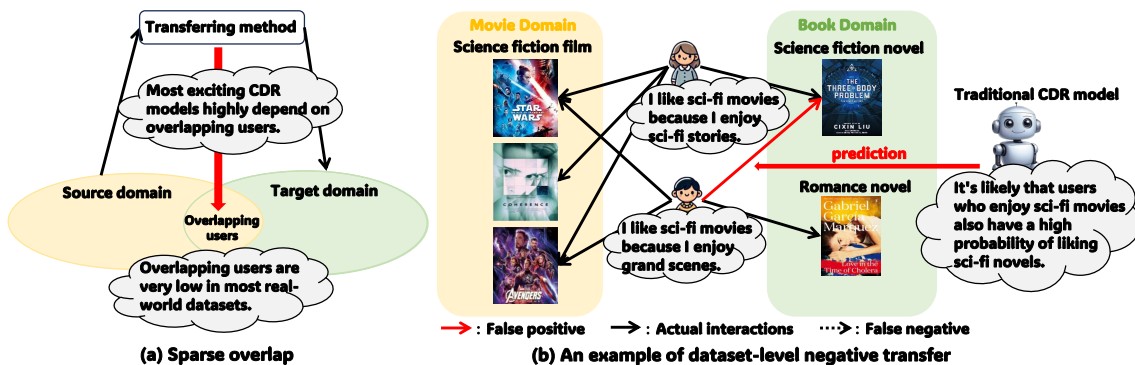

Figure 1: Illustration of challenges in Cross-Domain Recommendation (CDR). (a) Overlapping users' preferences for sci-fi movies (driven by grand scenes rather than narrative) lead to false positives (e.g., recommending sci-fi novels) and false negatives (e.g., missing romance novels) in the book domain when using traditional CDR models. (b) Most CDR models rely heavily on overlapping users to transfer information from the source domain to the target domain, but overlapping users are very low in most real-world datasets, making effective knowledge transfer difficult.

based on shared latent space Hu et al. (2018); Singh & Gordon (2008a) or transfer function Liu et al. (2020); Yang et al. (2024) to transfer information in different domains. These methods assume that users exhibit similar behavior patterns in different domains, allowing knowledge transfer through a shared representation.

However, there are still two open problems: **(i) Sparse overlap**, In real-world benchmarks like Amazon, the proportion of users active in both domains is often extremely low (e.g., below 5%), which severely restricts the ability of many state-of-the-art CDR models (e.g., Liu et al. (2020); Zhao et al. (2021); Li et al. (2024)) to establish effective cross-domain bridges. Existing approaches have aimed to address this challenge Wang et al. (2024); Xiang et al. (2025) by mainly considering how to efficiently utilize overlapping users at the model level to solve this problem. **(ii) Negative transfer**, which occurs when knowledge transferred from the source domain adversely affects performance in the target domain due to domain discrepancies or irrelevant information Cao et al. (2022a); Li et al. (2024; 2023). Recently, several methods have been proposed to effectively address negative transfer. In general, these approaches can be categorized into two main streams: (1) Filtering-based approaches, which selectively transfer highly relevant knowledge, such as user groups or interactions, while excluding irrelevant or noisy data to mitigate negative transfer Li et al. (2024); Song et al. (2024); (2) Disentanglement-based approaches, which explicitly separate domain-shared and domain-specific representations, thus preventing the propagation of domain-specific noise across domains Cao et al. (2022a; 2023).

Although recent work has made impressive progress in these two challenges of CDR separately, most efforts are model-centric methods that tacitly assume that user interactions are objective, unbiased, and recorded in perfectly clean logs Lai et al. (2024). However, many of these challenges are fundamentally dataset-level problems. In tackling sparse overlap, while many models aim to efficiently leverage overlapping users Wang et al. (2024); Xiang et al. (2025), as is shown in Fig 1(a) their performance is still bottlenecked by the sheer scarcity of these users in the data itself. Similarly, in addressing negative transfer, the issue often stems from the dataset level: noisy or non-causal interactions imported from the source domain can mislead the model and distort its understanding of the target domain's preferences. Consequently, even the most sophisticated model-centric approaches will be misled if they are trained on data that are inherently flawed for the transfer task. For example, as is shown in Fig 1(b), consider a scenario where the source domain involves movie interactions and the target domain involves book recommendations. A user shows strong preferences

for science fiction movies, driven primarily by grand visual effects rather than a genuine affinity for the science fiction narrative. Transferring this "science fiction"~~"science fiction"~~ genre preference directly to the target domain can lead to suboptimal outcomes, as it may result in recommending books with superficial thematic similarities that overlook users' actual interests in narrative depth. This introduces noisy shared representations, where the apparent preference stems from biased, superficial correlations rather than authentic causal links.

In order to fundamentally address both challenges, we propose a generate-filter dataset regeneration framework that operates entirely at the data level, enhancing the source dataset to better align with the target domain. To tackle sparse overlap, we first pretrain a self-supervised prediction model on the combined graph, masking interactions of selected overlapping users and reconstructing them alongside whole-graph predictions. This model is then frozen to generate high-confidence candidate interactions in the source domain for non-overlapping target users, effectively augmenting the overlap and bridging domain gaps. To mitigate negative transfer, inspired by causal inference and leveraging Structural Causal Models (SCM) as defined in Appendix 2, we introduce a counterfactual filtering process that observes changes in target-domain prediction performance. Specifically, we train a GNN to learn continuous edge weights and perform inverse optimization, retaining only those source interactions whose removal would degrade target accuracy—thus pruning non-causal, noisy edges prior to training. Although this edge impact assessment technique draws from prior work, such as Ma et al. (2022) for causal graph analysis, our regenerated dataset serves as a plug-and-play enhancement compatible with any CDR backbone, yielding tighter coupling and explicit causal connections between domains.

The main contributions of this work are as follows:

- We introduce a generate-filter dataset regeneration framework for CDR that operates at the data level, effectively addressing sparse overlap and negative transfer problems.

- We propose a self-supervised generation module to hallucinate plausible synthetic interactions for non-overlapping target-domain users in the source domain, thereby enhancing cross-domain connections.

- We develop a counterfactual filtering process to identify and retain causal interactions by observing their impact on target-domain prediction performance, mitigating biases and spurious correlations at their root.

- Our approach serves as a plug-and-play enhancement compatible with any CDR backbone. Extensive experiments on Douban and Amazon datasets demonstrate significant improvements, with gains up to 23.81% in Recall@10 and 22.22% in NDCG@10 across various baselines.

## 2 PRELIMINARIES

In this section, we introduce the task of this study and related concepts. First, we formally define Cross-Domain Recommendation.

**Definition 2.1** (Domain). *A domain is a triple $\mathbb{D} = (\mathcal{U}, \mathcal{I}, \mathcal{E})$, where $\mathcal{U}$ is the set of users, $\mathcal{I}$ is the set of items, $\mathcal{E} \subseteq \mathcal{U} \times \mathcal{I}$ is the set of observed interactions (edges).*

**Definition 2.2** (Feedback Space and Feedback Function). *Let $\mathbb{D} = (\mathcal{U}, \mathcal{I}, \mathcal{E})$ with $\mathcal{E} \subseteq \mathcal{U} \times \mathcal{I}$. The feedback space $\mathcal{Y}$ is the set of all possible user responses to items, such as binary clicks $\{0, 1\}$, discrete ratings $\{1, 2, 3, 4, 5\}$, or continuous scores. We define a feedback function*

$$l : \mathcal{E} \to \mathcal{Y},$$

*such that for each $(u, i) \in \mathcal{E}$, the value $l(u, i) \in \mathcal{Y}$ is the observed feedback of user $u$ on item $i$ in domain $\mathbb{D}$. For brevity, we denote $y_{ui} := l(u, i)$.*

**Definition 2.3** (Cross-Domain Recommendation (CDR) Task). *Given a changeable source domain $\mathbb{D}_S = (\mathcal{U}^S, \mathcal{I}^S, \mathcal{E}^S)$ with certain feedback function $l(\mathcal{E}^S)$, and a fixed target domain $\mathbb{D}_T = (\mathcal{U}^T, \mathcal{I}^T, \mathcal{E}^T)$, the goal of the CDR task is to learn a function*

$$\mathcal{F}_\theta^T(u, i \mid \mathbb{D}_S, l(\mathcal{E}^S)) : \mathcal{U}^T \times \mathcal{I}^T \mapsto \mathcal{Y},$$

*parameterized by $\theta$. CDR function leverages knowledge from both the source domain $\mathbb{D}_S$ and target domain $\mathbb{D}_T$ to predict the feedback of user $u \in \mathcal{U}^T$ on item $i \in \mathcal{I}^T$ in the target domain.*

Assume a non-empty overlap of users, defined as $\mathcal{U}^O := \mathcal{U}^S \cap \mathcal{U}^T \neq \varnothing$. Let $\mathcal{U}^{S'} := \mathcal{U}^S \setminus \mathcal{U}^O$ and $\mathcal{U}^{T'} := \mathcal{U}^T \setminus \mathcal{U}^O$ denote the non-overlapping users in the source and target domains, respectively. In the target domain $\mathbb{D}_T$, for each $u \in \mathcal{U}^T$ and $i \in \mathcal{I}^T$, with a binary feedback function $y_{u,i} \in \mathcal{Y} = \{0, 1\}$, we define the set of observed positive interactions as $\mathcal{E}_{\text{pos}}^T := \{(u, i) \mid y_{u,i} = 1\}$, and the set of observed negative interactions as $\mathcal{E}_{\text{neg}}^T := \{(u, i) \mid y_{u,i} = 0\}$.

However, while the CDR task, as defined above, focuses on learning a predictive function through model-level strategies, such approaches often overlook inherent data-level issues like biases and spurious correlations that persist across domains. These limitations highlight the need for interventions at the dataset level to complement and enhance model performance. Our proposed dataset-level solution addresses this by regenerating the source domain data to filter out non-causal interactions, offering a unique and effective supplement to existing CDR strategies. It is defined formally as follows:

**Definition 2.4** (Dataset regeneration for CDR). *Given the source domain dataset $\mathbb{D}_S = (\mathcal{U}^S, \mathcal{I}^S, \mathcal{E}^S)$ and the target domain dataset $\mathbb{D}_T = (\mathcal{U}^T, \mathcal{I}^T, \mathcal{E}^T)$. Dataset regeneration refers to learning a transformation function*

$$f : \mathbb{D}_S \times \mathbb{D}_T \mapsto \mathbb{D}_{S'} = (\mathcal{U}^S, \mathcal{I}^S, \mathcal{E}'),$$

*such that the regenerated dataset $\mathbb{D}_{S'}$ improves the utility of $\mathbb{D}_S$ for cross-domain recommendation tasks involving $\mathbb{D}_T$ (e.g., in terms of NDCG@K, Hit Rate@K and Recall@K on the target domain).*

While the concept of filtering or refining a dataset to mitigate negative transfer—where knowledge from the source domain ($\mathbb{D}_S$) adversely affects the target domain ($\mathbb{D}_T$)—is intuitive, systematically identifying which interactions in the source domain contribute to this issue poses a significant challenge. To address this, we turn to causal inference, specifically leveraging the framework of Structural Causal Models (SCM), which provides a rigorous method to evaluate how modifications in the source domain causally influence the target domain. This approach is particularly valuable in the context of dataset regeneration (as defined earlier), where the goal is to enhance cross-domain recommendation by removing detrimental interactions. In practice, our causal interventions focus on disrupting non-causal pathways from the source to the target domain to alleviate negative transfer. This involves selectively removing edges that do not contribute to accurate predictions, guided by a counterfactual evaluation process. To formalize this, we introduce the following definition:

**Definition 2.5** (Counterfactual Evaluation of Causal Interactions). *Given a subset of source domain interactions $\overline{\mathcal{E}^{S'}} \subseteq \mathcal{E}^S$ and a prediction function $\mathcal{F}_\theta^T(u, i \mid \mathbb{D}_S, l(\mathcal{E}^S))$, we assess the causal impact of these edges through counterfactual reasoning. This involves comparing the factual prediction $\mathcal{F}_\theta^T(u, i \mid \mathbb{D}_S, l(\mathcal{E}^S))$ with the counterfactual prediction under the intervention $do(\mathcal{E}^{S'} = \mathcal{E}^S \setminus \overline{\mathcal{E}^{S'}})$, where $do(\cdot)$ denotes Pearl's do-operator. If removing $\overline{\mathcal{E}^{S'}}$ results in unchanged or improved prediction performance in the target domain, $\overline{\mathcal{E}^{S'}}$ is classified as a spurious co-occurrence and marked for removal. Conversely, if the removal leads to a degradation in performance, the interaction is deemed causal and retained.*

This counterfactual evaluation serves as the foundation for our dataset regeneration strategy, enabling us to distinguish between causal and spurious interactions with precision. By systematically applying this process, we can construct a refined source dataset that enhances the robustness of cross-domain recommendations,

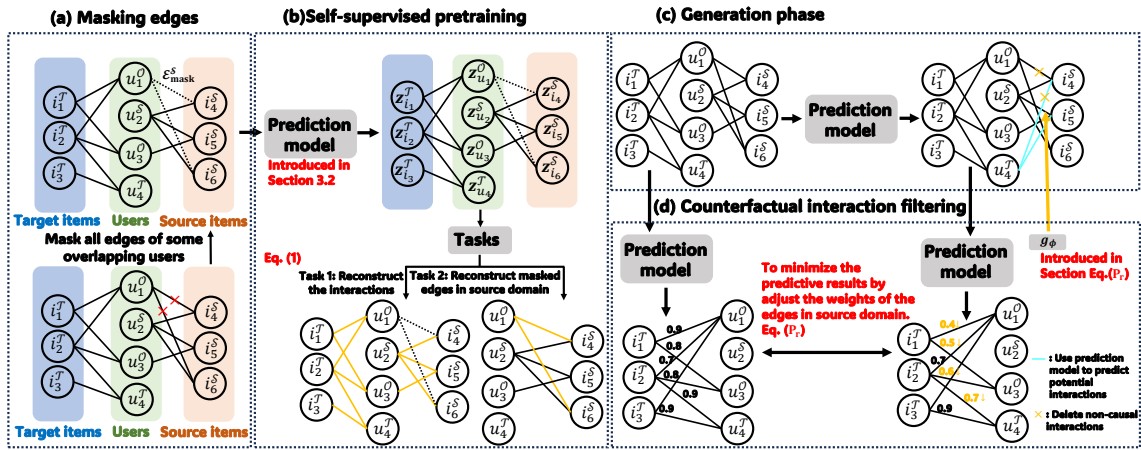

Figure 2: The overall pipeline of our proposed method is illustrated in this figure. We first perform a pretraining process, during which we randomly mask all edges of some selected users. The model is then trained through two tasks simultaneously. Next, we fix the parameters of the prediction model. Afterwards, we predict the most likely interactions in the source domain for the target-specific users in the target domain, obtaining candidate edges for generation. Finally, we utilize $g_\phi$ to adjust the weights of edges in the source domain, aiming to maximally reduce the performance of the prediction model in the target domain.

particularly in scenarios with sparse overlapping users. ~~The next section details the optimization techniques used to implement this filtering, integrating the pre-trained prediction model with edge weight adjustments to achieve the desired causal pruning.~~

## 3 METHODOLOGY

### 3.1 OVERALL STRUCTURE

In this section, we outline our method as a unified workflow, which is shown in Fig 2. We begin by pre-training a prediction model, $\mathcal{F}_\theta^T$, which serves as a surrogate for subsequent steps, including generating interactions for non-overlapping users in the source domain and filtering causal edges. Due to space constraints, we have placed the detailed structure of the prediction model $\mathcal{F}_\theta^T$ in the Appendix B. Next, we freeze the parameters of this pre-trained model and predict the most likely interactions for non-overlapping users $u \in \mathcal{U}^{T'} := \mathcal{U}^T \setminus \mathcal{U}^O$ with items $i^S \in \mathcal{I}^S$ in the source domain, selecting these as candidate synthetic edges. We then train another GNN model to predict which edges, when removed, cause the greatest degradation in the pre-trained model's performance on the target domain, thereby identifying edges with causal relevance to the target domain.

### 3.2 OVERLAPPING USERS GENERATION

Most existing CDR approaches rely on a large set of overlapping users to transfer knowledge through shared user representations. However, real-world datasets (e.g., Amazon) often contain only a few shared users, which significantly restricts the ability of these models to transfer knowledge across domains and thus hampers their performance on real-world applications. Moreover, in the CDR setting, many users appear in only one domain, and their absence in the other domain is not necessarily due to a lack of interest, but rather because they are unaware of the existence of that domain. As a result, the model fails to capture the potential connections that should exist between the two domains. To address this issue, we let the model learn from the

behavior patterns of existing overlapping users of both domains and then predict the potential behaviors of users who have no interactions in one of the domains. To achieve this, we first pre-train a prediction model, which is detailed in Appendix B, and then use it to predict these possible interactions.

**Self-supervised pretraining.** We randomly mask all interactions of a number of overlapping users in the source domain, and we denote these interactions as $\mathcal{E}_{\text{mask}}^S$. Then, by learning their behaviors in the target domain as well as the behaviors of other users, the model aimed at reconstructing these masked interactions. In order to learn information from the entire graph, we also train the model to predict the remaining interactions in both the source and target domains. Concretely, we optimize the following BPR objective:

$$\mathcal{L} = \underbrace{\sum_{(u,i)\in\mathcal{E}_{\text{pos}}^S\cup\mathcal{E}_{\text{pos}}^T}\sum_{(u,j)\in\mathcal{E}_{\text{neg}}^S\cup\mathcal{E}_{\text{neg}}^T}\log\sigma\big(\hat{y}_{u,i}-\hat{y}_{u,j}\big)}_{\text{Whole-graph interaction prediction}} + \underbrace{\sum_{(u,i^S)\in\mathcal{E}_{\text{mask}}^S}\sum_{(u,j^S)\in\mathcal{E}_{\text{neg}}^S}\log\sigma\big(\hat{y}_{u,i^S}-\hat{y}_{u,j^S}\big)}_{\text{Masked-edge reconstruction}}, \tag{1}$$

where $j^T$ and $j^S$ are negative samples from $\mathcal{I}^T$ and $\mathcal{I}^S$, respectively.

**Generation phase.** After fine-tuning, we freeze the surrogate $\mathcal{F}_\theta$ and, for every non-overlapping user $u \in \mathcal{U}^{T'}$, score all candidate pairs $(u, i^S)$ with $i^S \in \mathcal{I}^S$ to obtain $\hat{y}_{u,i^S}$. We then select the $k$ highest-scoring items and define the synthetic set $\widehat{\mathcal{E}}^S := \big\{(u, i^S)\,\big|\,i^S \text{ is in the top-}k\big\}$. The augmented edge set is $\widetilde{\mathcal{E}}^S := \mathcal{E}^S \cup \widehat{\mathcal{E}}^S$, yielding the dataset

$$\widetilde{\mathbb{D}}_S = \big(\mathcal{U}^S\cup\mathcal{U}^T,\ \mathcal{I}^S,\ \widetilde{\mathcal{E}}^S\big),$$

which connects every target-domain user to the source domain and serves as the input for the subsequent counterfactual filtering described in the next section.

### 3.3 COUNTERFACTUAL INTERACTION FILTERING

Based on Definition 2.5, our aim is to identify which part of the interactions in the source domain has a causal effect on recommendations in the target domain. To pinpoint the source-domain interactions that truly affect the target domain, we pose the following counterfactual question: *'If the recommender system did not observe certain interactions in the source domain, would it still provide accurate predictions in the target domain?'* This question guides us to distinguish causal interactions from spurious co-occurrences.

An intuitive strategy is to identify non-causal edges by optimizing the removal of interactions so as to maximize the prediction model's performance on the target domain $\mathcal{E}_{\text{pos}}^T$, and it can be formalized into an optimization problem.

Suppose $|\mathcal{E}^S| = N$, the feedback function is defined as a binary function, and $\mathbf{y}_S := \big(l(e)\big)_{e\in\mathcal{E}^S} \in \{0,1\}^N$.

$$\max_{\mathcal{E}^{S'}\subseteq\mathcal{E}^S}\quad \mathcal{J}(\mathcal{E}^{S'}) = \sum_{(u,i)\in\mathcal{E}^T}\Big[\mathcal{F}_\theta^T\big(u,i\mid\mathbb{D}_S,l(\mathcal{E}^S)\big)-\mathcal{F}_\theta^T\big(u,i\mid\mathbb{D}_{S'},l(\mathcal{E}^{S'})\big)\Big] \tag{P}$$

$$\text{s.t.}\qquad N_{\max}\geq|\mathcal{E}_{\text{neg}}^{S'}|,\quad \mathbf{y}_S,\ \mathbf{y}_{S'}\in\{0,1\}^N,$$

where $N_{\max}$ denotes the maximum number of negative edges in $\mathcal{E}_{\text{neg}}^{S'}$ that are allowed to be retained, and $N_{\max}$ is used as a threshold in the regularization term for negative edge.

However, directly solving the following often leads to overfitting: the optimization may exploit spurious patterns in the training data, failing to generalize and improve genuine target-domain predictions. Instead, we adopt an inverse approach inspired by adversarial robustness techniques. We optimize to find edges whose removal maximally degrades the surrogate model's performance. These edges are deemed causal, as their presence is critical for accurate predictions; the remaining edges are non-causal and can be filtered out to mitigate negative transfer.

For the fixed domain $\mathbb{D}$ and feedback function $l$, we have the following formula:

$$\sum_{(u,i)\in\mathcal{E}^T} \mathcal{F}_\theta^T = \sum_{(u,i)\in\mathcal{E}_{pos}^T} \mathcal{F}_\theta^T - \sum_{(u,i)\in\mathcal{E}_{neg}^T} \mathcal{F}_\theta^T .$$

Since the variable of problem (P) is the subset $\mathcal{E}^{S'} \subseteq \mathcal{E}^S$, then $\sum_{(u,i)\in\mathcal{E}^T} \mathcal{F}_\theta^T\big(u,i \mid \mathbb{D}_S, l(\mathcal{E}^S)\big)$ is a constant value for fixed parameter $\theta$, and the objective function of the problem (P) is equivalent to minimizing $\sum_{(u,i)\in\mathcal{E}^T} \mathcal{F}_\theta^T\big(u,i \mid \mathbb{D}_S, l(\mathcal{E}^{S'})\big)$. We have $N_{\max} \geq |\mathcal{E}_{neg}^{S'}| = \|\mathbf{1} - \mathbf{y}_{S'}\|_0$, so the exact-penalty objective function of problem (P) is

$$\mathcal{J}(\mathcal{E}^{S'}) - \lambda\left[\|\mathbf{1} - \mathbf{y}_{S'}\|_0 - N_{\max}\right]_+ ,$$

where $[x]_+ := \max(0, x)$, $\lambda > 0$ .

Formally, this leads to the following optimization problem:

$$\min_{\mathcal{E}^{S'}\subseteq\mathcal{E}^S} \overline{\mathcal{J}}(\mathcal{E}^{S'}) = \left[ \sum_{(u,i)\in\mathcal{E}'_{\text{pos}}} \mathcal{F}_\theta^T\big(u,i \mid \mathbb{D}_{S'}, l(\mathcal{E}^{S'})\big) - \sum_{(u,j)\in\mathcal{E}'_{\text{neg}}} \mathcal{F}_\theta^T\big(u,j \mid \mathbb{D}_{S'}, l(\mathcal{E}^{S'})\big) + \lambda\left[\|\mathbf{1} - \mathbf{y}_{S'}\|_0 - N_{\max}\right]_+ \right]$$

$$\text{s.t.} \qquad\qquad \mathbf{y}_{S'} \in \{0,1\}^N, \tag{$\overline{\text{P}}$}$$

Worse still, solving this discrete edge-selection problem is intractable, as it constitutes an NP-hard combinatorial optimization , which is formally given by the following lemma:

**Lemma 1.** *Let* $(\text{P}_{\text{cov}})$ *denote the Maximum Coverage problem. The decision version* $(\text{P}_{\text{cov}})_d$*, which is known to be NP-complete, reduces in polynomial time to the decision version of Problem (P), denoted as* $(\text{P})_d$*:*

$$(\text{P}_{\text{cov}})_d \leq_p (\text{P})_d.$$

*Furthermore, Problem $(\overline{\text{P}})$ is polynomial-time equivalent to Problem (P)*

$$(\text{P}) \equiv_p (\overline{\text{P}}).$$

*Consequently, Problem $(\overline{\text{P}})$ is NP-hard.*

The proof is given in Appendix D.

To make it differentiable, we relax the binary (0/1) edge inclusion decisions into continuous edge weights. Specifically, we introduce a trainable feedback function $g_\phi : \mathcal{E}^S \to [0, 1]$, implemented by any GNN models.

The pre-trained prediction model $\mathcal{F}_\theta^T$ is adapted to incorporate these weights into its aggregation process, which is defined as $\mathbf{w}_S := (l_1(e))_{e\in\mathcal{E}^S} = \big(\sigma(g_\phi(e))\big)_{e\in\mathcal{E}^S} \in (0,1)^N$, and also $\mathbf{y}_S := \big(l_2(e)\big)_{e\in\mathcal{E}^S} \in \{0,1\}^N$. Then, the relaxed differentiable objective is:

$$\min_\phi \mathcal{L}(\phi; \mathcal{E}^{S'}) = \left[ \sum_{(u,i)\in\mathcal{E}'_{\text{pos}}} \mathcal{F}_\theta^T\big(u,i \mid \mathbb{D}_{S'}, l_1(\mathcal{E}^{S'})\big) - \sum_{(u,j)\in\mathcal{E}'_{\text{neg}}} \mathcal{F}_\theta^T\big(u,j \mid \mathbb{D}_{S'}, l_1(\mathcal{E}^{S'})\big) + \lambda\|\mathbf{w}_{S'} \odot (\mathbf{1} - \mathbf{y}_{S'})\|_1 \right]$$

$$\text{s.t.} \qquad\qquad \mathbf{w}_{S'} \in (0,1)^N, \tag{$\text{P}_{\text{r}}$}$$

where $\odot$ denotes the Hadamard product, the $\ell_1$ regularization term with coefficient $\lambda > 0$ penalizes weights of negative edges, encouraging most weights to remain high (close to 1) while allowing only a few to be lowered to achieve performance degradation.

After optimization, interactions with low weights ($w_e < \tau$) are classified as causal, since attenuating them significantly worsens predictions. To regenerate the dataset, we retain these causal edges in $\mathcal{E}'$ for the filtered source domain $\mathbb{D}_{S'}$, discarding the non-causal ones ($w_e \geq \tau$) that contribute to negative transfer.

The relationships among the three optimization problems ($\overline{\text{P}}$, P, and $\text{P}_{\text{r}}$) are given by the following lemma:

**Lemma 2.** *Problem ($\overline{\mathrm{P}}$) is equivalent to problem (P), and problem ($\mathrm{P_r}$) provides relaxed solution:*

$$\mathcal{L}(\phi^*; \mathcal{E}^{S'}) - \lambda N_{max} \leq \overline{\mathcal{J}}((\mathcal{E}^{S'})^*)$$

*where $(\cdot)^*$ denotes the optimal solution, and $\lambda > 0$ is certain penalty parameter, $N \geq N_{max} \geq 0$.*

The proof is given in Appendix D.

## 4 EXPERIMENTS

In this section, we conduct extensive experiments on multiple public datasets to evaluate the proposed method.

### 4.1 SETUP

**Datasets.** To assess the effectiveness of our proposed approach alongside various baseline models, we utilize three distinct domain pairs sourced from two widely recognized real-world cross-domain datasets, Douban datasets and Amazon datasets. And we designed six cross-domain recommendation tasks. Each domain within a pair is alternately designated as the target domain for evaluation. To facilitate the reproduction of our results, we have included the specific data, processing methods, and download sources of the dataset in ~~the~~ Appendix E.1.

**Baselines.** We compared our methods with two single-domain methods (1) **MF** Koren et al. (2009) (2) **Light-GCN** He et al. (2020) and six cross domain methods (3) **CMF** Singh & Gordon (2008b) (4) **EMCDR** Man et al. (2017) (5) **BiTGCF** Liu et al. (2020) (6) **DTCDR** Zhu et al. (2019) (7) **CAT-ART** Li et al. (2023) (8) **UniCDR** Cao et al. (2023) (9) **CUT** Li et al. (2024). The single-domain baselines, trained exclusively on the target dataset and the cross domain baselines are trained on both domains and tested on the target domain. To ensure the fairness of the comparison, we adopted the Recbole-CDRZhao et al. (2022) framework for all baselines. And more detailed information on these methods and implementations is provided in Appendix E.3.

**Evaluation metrics.** Evaluations are performed using a full ranking approach, considering all items in the datasets. All models are evaluated by Recall@K and NDCG@K, where $K$ is set to 10 in this research. The formal definition of their metrics is detailed in Appendix E.2

**Experimental environment.** All experiments are executed on a system featuring an NVIDIA GeForce RTX 3090 GPU and an Intel Core i7-13700F CPU. The implementation leverages the RecBole-CDRZhao et al. (2022) toolkit, offering a consistent platform for cross-domain recommendation studies.

### 4.2 OVERALL PERFORMANCE

To evaluate the overall effectiveness and generalizability of our proposed Gen/Del modules, we integrated them with three distinct backbones: LightGCN, CUT, and BiTGCF. We then benchmarked these enhanced models against a comprehensive suite of single-domain and cross-domain baselines. The full results, presented in Table 1, demonstrate that our method consistently achieves state-of-the-art performance across various datasets.

1. Against strong baselines such as EMCDR, DTCDR, CAT-ART, UniCDR, BiTGCF, and CUT, our methods deliver consistent gains. Notably, CUT+Gen/Del attains the highest scores in Sport→Cloth (0.0546 in R, 0.0308 in N) and Music→Movie (0.1396 in R, 0.1442 in N), with statistical significance ($p < 0.05$) over the best baselines, highlighting the modules' ability to alleviate negative transfer.

2. On Amazon datasets, our methods excel in asymmetric pairs like Cloth→Video and Video→Cloth, with BiTGCF+Gen/Del leading at 0.1389 in R and 0.0766 in N for Cloth→Video. On Douban,

Table 1: Performance comparison of three method categories: single-domain, cross-domain, and our proposed method applied to three representative backbones. For the experiment of LightGCN+Gen/Del, as LightGCN is a single-domain baseline, we merge the source and target data for training, and evaluation was performed on the target domain. **Bolded** and underlined values indicate the best and second-best results in each column, respectively. * denotes that our method achieves statistically significant improvement over the best baseline in that column (paired t-test with p-value $< 0.05$).

| | | Amazon | | | | | | | | Douban | | | |
| --- | --- | --- | --- | --- | --- | --- | --- | --- | --- | --- | --- | --- | --- |
| | | Cloth→Sport | | Sport→Cloth | | Cloth→Video | | Video→Cloth | | Movie→Music | | Music→Movie | |
| | Method | R | N | R | N | R | N | R | N | R | N | R | N |
| Single domain | MF | 0.0492 | 0.0270 | 0.0243 | 0.0137 | 0.1153 | 0.0623 | 0.0243 | 0.0137 | 0.1004 | 0.0733 | 0.1053 | 0.0997 |
| | LightGCN | 0.0604 | 0.0331 | 0.0385 | 0.0207 | 0.1181 | 0.0639 | 0.0385 | 0.0207 | 0.1069 | 0.0806 | 0.1031 | 0.1096 |
| Cross domain | CMF | 0.0545 | 0.0293 | 0.0291 | 0.0157 | 0.1194 | 0.0644 | 0.0246 | 0.0136 | 0.0944 | 0.0725 | 0.0946 | 0.1031 |
| | EMCDR | 0.0538 | 0.0288 | 0.0234 | 0.0127 | 0.1165 | 0.0633 | 0.0232 | 0.0124 | 0.1014 | 0.0756 | 0.1064 | 0.1156 |
| | DTCDR | 0.0558 | 0.0332 | 0.0263 | 0.0141 | 0.1085 | 0.0584 | 0.0241 | 0.0126 | 0.0881 | 0.0658 | 0.0943 | 0.0982 |
| | CAT-ART | 0.0515 | 0.0276 | 0.0240 | 0.0130 | 0.1133 | 0.0609 | 0.0245 | 0.0123 | 0.0901 | 0.0685 | 0.1055 | 0.1048 |
| | UniCDR | 0.0624 | 0.0340 | 0.0433 | 0.0239 | 0.1249 | 0.0684 | 0.0349 | 0.0191 | 0.1073 | 0.0754 | 0.1095 | 0.0994 |
| | BiTGCF | 0.0655 | 0.0360 | 0.0508 | 0.0286 | 0.1311 | 0.0715 | 0.0411 | 0.0230 | 0.1228 | 0.0918 | 0.1289 | 0.1219 |
| | CUT | 0.0653 | 0.0364 | 0.0441 | 0.0252 | 0.1303 | 0.0720 | 0.0381 | 0.0213 | 0.1205 | 0.0946 | 0.1393 | 0.1437 |
| Our methods | LightGCN+Gen/Del | 0.0632 | 0.0354 | 0.0458 | 0.0262 | 0.1344 | 0.0733 | 0.0422 | 0.0236 | 0.1097 | 0.0785 | 0.1024 | 0.1008 |
| | CUT+Gen/Del | 0.0652 | 0.0366 | **0.0546*** | **0.0308*** | 0.1325 | 0.0740 | 0.0463 | 0.0256 | **0.1248*** | **0.0962*** | **0.1396*** | **0.1442*** |
| | BiTGCF+Gen/Del | **0.0692*** | **0.0378*** | 0.0532 | 0.0297 | **0.1389*** | **0.0766*** | **0.0498*** | **0.0278*** | 0.1237 | 0.0926 | 0.1362 | 0.1275 |

CUT+Gen/Del dominates Movie→Music and Music→Movie. Overall, our approach yields average improvements of 5-20% over the strongest baselines, validating its robustness and adaptability.

Table 2: Ablation study of the proposed generative modules on the BiTGCF backbone. Each cell for a variant shows the absolute metric value, with the relative improvement over the vanilla BiTGCF in parentheses. Improvements are in orange, drops in cyan, with color intensity scaling with the magnitude of change.

| Dataset | BiTGCF | | BiTGCF+Del | | BiTGCF+Gen | | BiTGCF+Gen/Del | |
| --- | --- | --- | --- | --- | --- | --- | --- | --- |
| | R | N | R | N | R | N | R | N |
| Cloth→Sport | 0.0655 | 0.0360 | 0.0672 (+2.60%) | 0.0378 (+5.00%) | 0.0668 (+1.98%) | 0.0374 (+3.89%) | 0.0692 (+5.65%) | 0.0378 (+5.00%) |
| Sport→Cloth | 0.0508 | 0.0286 | 0.0523 (+2.95%) | 0.0292 (+2.10%) | 0.0513 (+0.98%) | 0.0293 (+2.45%) | 0.0532 (+4.72%) | 0.0297 (+3.85%) |
| Cloth→Video | 0.1311 | 0.0715 | 0.1392 (+6.18%) | 0.0768 (+7.41%) | 0.1297 (-1.07%) | 0.0703 (-1.68%) | 0.1389 (+5.95%) | 0.0766 (+7.13%) |
| Video→Cloth | 0.0411 | 0.0230 | 0.0462 (+12.41%) | 0.0256 (+11.30%) | 0.0458 (+11.44%) | 0.0252 (+9.57%) | 0.0498 (+21.17%) | 0.0278 (+20.87%) |
| Movie→Music | 0.1209 | 0.0906 | 0.1216 (+0.58%) | 0.0918 (+1.32%) | 0.1212 (+0.25%) | 0.0911 (+0.55%) | 0.1237 (+2.32%) | 0.0926 (+2.21%) |
| Music→Movie | 0.1289 | 0.1219 | 0.1344 (+4.27%) | 0.1271 (+4.27%) | 0.1320 (+2.41%) | 0.1262 (+3.53%) | 0.1362 (+5.66%) | 0.1275 (+4.59%) |

## 4.3 ABLATION STUDY

To validate the individual contributions of our proposed generative modules, we conduct a detailed ablation study. Specifically, we investigate the impact of the deletion-based module (**+Del**) and the generation-based module (**+Gen**) by applying them both individually and in combination (**+Gen/Del**) to one of our backbones, BiTGCF. The results, summarized in Table 2. Our key findings are as follows:

1. The Del module alone yields consistent improvements across most datasets by removing noisy or spurious interactions. For instance, it achieves **+12.41%** in Recall@10 and **+11.30%** in NDCG@10 on Video→Cloth, effectively mitigating negative transfer in asymmetric domain pairs.

2. The +Gen module individually enhances performance in several cases, such as **+11.44%** in Recall@10 on Video→Cloth, by creating aligned interactions. However, it occasionally leads to minor drops (e.g., **-1.07%** in Recall@10 on Cloth→Video), likely due to imperfect generation without noise control.

3. Combining both modules results in the strongest gains, outperforming individual variants. Notable improvements include **+21.17%** in Recall@10 and **+20.87%** in NDCG@10 on Video→Cloth, demonstrating their complementary nature in generating useful data while deleting harmful noise for robust CDR.

## 5  CONCLUSION

We have introduced a dataset regeneration framework for cross-domain recommendation that addresses fundamental data sparsity challenges by generating synthetic interactions tailored to the target domain. By integrating a customized surrogate model for graph-based aggregation and causal-inspired techniques to reduce negative transfer, our method demonstrates superior performance across multiple datasets and CDR baselines. Experimental results validate its effectiveness, showing significant improvements in recommendation accuracy and robustness. While our approach advances data-centric CDR, limitations such as computational overhead in large-scale regeneration and potential biases in generated data warrant further exploration. Future work could extend this to multi-domain scenarios or incorporate advanced generative models like diffusion processes for even finer-grained data reproduction.

## 6  ETHIC STATEMENT

This work relies on publicly available datasets (Amazon and Douban) that contain anonymized user-item interactions. No new data collection was performed, and all datasets were used in accordance with their original terms of use and licensing. We acknowledge potential risks associated with our dataset regeneration framework. By generating synthetic interactions and filtering based on causal inference, the method could inadvertently amplify existing biases in the source data, such as demographic or cultural stereotypes (e.g., genre preferences tied to user groups), leading to unfair recommendations in the target domain. This might exacerbate issues like echo chambers or discriminatory outcomes in real-world recommender systems. To mitigate this, we encourage downstream users to evaluate regenerated datasets for fairness metrics (e.g., demographic parity) and apply debiasing techniques. Additionally, while our approach aims to reduce negative transfer, misuse could introduce spurious correlations if applied to sensitive domains like healthcare or finance, potentially causing harm through inaccurate predictions.

## 7  REPRODUCIBILITY STATEMENT

We detail our experimental setup, including datasets and baseline models, in Section 4.1. The source code for this work is fully available in the Supplementary Material and can also be accessed via the following anonymous link https://anonymous.4open.science/r/Dataset-regeneration-for-cross-domian-recommendation-D8FD/ for review purposes.

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

# Appendix

## A RELATED WORK

### A.1 CROSS-DOMAIN RECOMMENDATION

Cross-domain recommendation (CDR) addresses data sparsity by transferring knowledge across domains. Early approaches focused on **shared representations**, with CMF Singh & Gordon (2008a), CLFM Gao et al. (2013), and CoNet Hu et al. (2018) enabling parameter sharing between domains through matrix factorization or neural networks. However, these methods assume universal preference similarity across domains, which often fails in practice. Recent work has shifted toward **embedding alignment** strategies. BiTGCF Liu et al. (2020) employs transfer functions to align user embeddings, while CCDR Xie et al. (2022) and HCTS Yang et al. (2024) leverage contrastive learning to pull same-user embeddings closer across domains. Another prominent direction uses **mapping functions**, pioneered by EMCDR Man et al. (2017), with subsequent work Cao et al. (2023); Kang et al. (2019); Zhu et al. (2021) refining cross-domain correlations through learned transformations. Despite these advances, negative transfer remains a critical challenge. Recent methods address this through model-level constraints: DisenCDR Cao et al. (2022a) disentangles domain-shared and domain-specific representations; CDRIB Cao et al. (2022b) applies information bottleneck principles; UniCDR Cao et al. (2023) combines contrastive learning with domain masking. While effective, these approaches focus solely on model architectures, overlooking data-level issues that fundamentally cause negative transfer.

### A.2 DATASET REGENERATION FOR RECOMMENDER SYSTEMS

The emergence of data-centric recommender systems Lai et al. (2024) recognizes that model complexity alone cannot overcome inherent data limitations. This paradigm shift has spawned several research directions. **Denoising methods** tackle noisy interactions: ADT Wang et al. (2021) adaptively prunes noisy feedback during training; SGDL Gao et al. (2022) leverages early-training memorization patterns; SLED Zhang et al. (2023) employs structure learning for systematic denoising. For **incomplete data**, fairAC Guo et al. (2023) addresses missing attributes in graph learning, while You et al. (2020) proposes graph-based imputation. **Debiasing approaches** Schnabel et al. (2016); Saito et al. (2020) correct for selection and exposure biases through causal inference techniques. Recent work has explored data augmentation for sequential recommendations Yin et al. (2024). However, no prior research has applied dataset regeneration specifically to CDR scenarios. Our approach uniquely modifies source domain data to enhance cross-domain transfer, complementing existing single-domain methods.

### A.3 CAUSAL INFERENCE IN RECOMMENDER SYSTEMS

Causal inference has revolutionized recommender systems by moving beyond correlations to understand true causal effects, addressing fundamental issues like bias and robustness. **Debiasing and unbiased learning** represents the most mature application. Schnabel et al. Schnabel et al. (2016) introduced inverse propensity scoring (IPS) for recommendation, treating items as interventions. Recent advances include doubly robust estimators Wang et al. (2019), AutoDebias Chen et al. (2021) for automatic bias discovery, and CausInt Wang et al. (2022a) for handling multiple biases simultaneously. **Causal modeling and intervention** methods explicitly model recommendation processes through causal graphs. DICE Zheng et al. (2021) disentangles user interest from social conformity; CauseRec Zhang et al. (2021) removes popularity bias through backdoor adjustment; recent work explores causal prompting for LLM-based recommendations Zhang et al. (2025). These approaches intervene during model training to align with presumed causal structures. **Causal representation learning** seeks invariant features across environments. InvPref Wang et al. (2022b) learns stable preferences under distribution shifts; CausPref He et al. (2022) discovers causal preference structures for out-of-distribution generalization; DCCL Zhao et al. (2023) achieves causal disentanglement

via contrastive learning. Critically, all existing causal methods in recommender systems operate at the *single-domain, model level*. Our work pioneers a fundamentally different approach: **data-level causal intervention for cross-domain scenarios**. Rather than modifying learning algorithms, we directly regenerate the source dataset through counterfactual reasoning, identifying and filtering interactions that causally harm target domain performance. This data-centric causal filtering uniquely addresses cross-domain negative transfer at its root, offering a novel complement to model-centric solutions.

## B  DETAILED STRUCTURE OF THE PREDICTION MODEL

We adopted a structure similar to LightGCN He et al. (2020), but to accommodate the subsequent filtering process, we modified LightGCN's aggregation process to a weighted average that considers edge weights. In the $l$-th layer, we perform weighted average aggregation, where each neighbor's contribution is normalized by the sum of edge weights from the node:

$$\mathbf{e}_u^{(l)} = \sum_{i \in \mathcal{N}(u)} \frac{w_{u,i}}{\sum_{j \in \mathcal{N}(u)} w_{u,j}} \mathbf{e}_i^{(l-1)},$$

$$\mathbf{e}_i^{(l)} = \sum_{u \in \mathcal{N}(i)} \frac{w_{u,i}}{\sum_{v \in \mathcal{N}(i)} w_{v,i}} \mathbf{e}_u^{(l-1)},$$

where $\mathcal{N}(u)$ and $\mathcal{N}(i)$ are the neighbor sets, and $w_{u,i}$ is the edge weight between $u$ and $i$ and $\mathbf{e}_u^{(l)}$ and $\mathbf{e}_i^{(l)}$ are trainable embeddings of users and items. During the pre-training process, we set the edge weight as a fixed value of 1, which is the same as LightGCN. The final embeddings are averaged across $L$ layers:

$$\mathbf{e}_u = \frac{1}{L+1} \sum_{l=0}^{L} \mathbf{e}_u^{(l)}, \quad \mathbf{e}_i = \frac{1}{L+1} \sum_{l=0}^{L} \mathbf{e}_i^{(l)}.$$

The prediction score for a pair $(u, i)$ is the inner product:

$$\hat{y}_{u,i} = \mathbf{e}_u^\top \mathbf{e}_i.$$

For brevity, we denote this weighted-aggregation model by $\mathcal{F}_\theta$ in the following sections; when it is used solely to predict interactions in the target domain (as specified in Definition 2.3), we write it as $\mathcal{F}_\theta^T$.

## C  NOTATIONS

We first summarize in the following tables the notation used in the main text and in the subsequent detailed proofs to improve readability:

Table 3: Notation for datasets, graphs, and feedback.

| Symbol | Description |
| --- | --- |
| $\mathbb{D} = (\mathcal{U}, \mathcal{I}, \mathcal{E})$ | A recommendation domain: user set $\mathcal{U}$, item set $\mathcal{I}$, and observed edges $\mathcal{E} \subseteq \mathcal{U} \times \mathcal{I}$. |
| $\mathcal{Y}$ | Feedback space (e.g., $\{0, 1\}$ for implicit feedback, or rating scores). |
| $l : \mathcal{E} \to \mathcal{Y}$ | Feedback function; $l(u, i)$ is the observed feedback on edge $(u, i)$. |
| | Continued on next page |

| Symbol | Description |
|---|---|
| $y_{ui}$ | Shorthand for $l(u, i)$, the feedback of user $u$ on item $i$. |
| $\mathbb{D}_S = (\mathcal{U}^S, \mathcal{I}^S, \mathcal{E}^S)$ | Source-domain dataset: source users $\mathcal{U}^S$, source items $\mathcal{I}^S$, and source edges $\mathcal{E}^S$. |
| $\mathbb{D}_T = (\mathcal{U}^T, \mathcal{I}^T, \mathcal{E}^T)$ | Target-domain dataset: target users $\mathcal{U}^T$, target items $\mathcal{I}^T$, and target edges $\mathcal{E}^T$. |
| $\mathcal{U}^O := \mathcal{U}^S \cap \mathcal{U}^T$ | Overlapping users that appear in both source and target domains. |
| $\mathcal{U}^{S'} := \mathcal{U}^S \setminus \mathcal{U}^O$ | Non-overlapping source users (only in the source domain). |
| $\mathcal{U}^{T'} := \mathcal{U}^T \setminus \mathcal{U}^O$ | Non-overlapping target users (only in the target domain). |
| $\mathcal{E}^T_{\text{pos}}$ | Positive target-domain edges, e.g., $\mathcal{E}^T_{\text{pos}} = \{(u, i) \in \mathcal{E}^T \mid y_{ui} = 1\}$. |
| $\mathcal{E}^T_{\text{neg}}$ | Negative target-domain edges, e.g., $\mathcal{E}^T_{\text{neg}} = \{(u, i) \in \mathcal{E}^T \mid y_{ui} = 0\}$. |
| $\mathcal{E}^S_{\text{pos}}, \mathcal{E}^S_{\text{neg}}$ | Positive / negative edges in the source domain according to the binary feedback. |
| $f : \mathbb{D}_S \times \mathbb{D}_T \mapsto \mathbb{D}_{S'}$ | Dataset-regeneration mapping that transforms $(\mathbb{D}_S, \mathbb{D}_T)$ into a regenerated source dataset $\mathbb{D}_{S'}$. |
| $\mathbb{D}_{S'} = (\mathcal{U}^S, \mathcal{I}^S, \mathcal{E}')$ | Regenerated source-domain dataset; shares $\mathcal{U}^S$ and $\mathcal{I}^S$, but has updated edge set $\mathcal{E}'$. |
| $\mathcal{E}^S_{\text{mask}}$ | Masked source-domain edges used as reconstruction targets in self-supervised pretraining. |
| $\hat{\mathcal{E}}^S$ | Synthetic source edges generated for non-overlapping target users (e.g., top-$k$ predicted items). |
| $\tilde{\mathcal{E}}^S := \mathcal{E}^S \cup \hat{\mathcal{E}}^S$ | Augmented source edge set that combines original and synthetic edges. |
| $\tilde{\mathbb{D}}_S$ | Augmented source graph built from $(\mathcal{U}^S \cup \mathcal{U}^T, \mathcal{I}^S, \tilde{\mathcal{E}}^S)$. |

Table 4: Notation for the prediction model and optimization problems (P), ($\bar{\text{P}}$) and (P$^r$).

| Symbol | Description |
|---|---|
| $F_\theta$ | GNN-based prediction model (e.g., LightGCN-style) with parameters $\theta$. |
| $F_\theta^T$ | Target-domain prediction function induced by $F_\theta$. |
| $N(u), N(i)$ | Neighbor sets in the bipartite graph: items linked to user $u$, or users linked to item $i$. |
| $w_{u,i}$ | Edge weight between user $u$ and item $i$ in the GNN aggregation. |
| $e_u^{(l)}, e_i^{(l)}$ | Embeddings of user $u$ and item $i$ at GNN layer $l$. |
| $L$ | Number of GNN layers in $F_\theta$. |
| $e_u, e_i$ | Final user and item embeddings obtained by layer-wise averaging. |

| Symbol | Description |
|---|---|
| $\hat{y}_{u,i}$ | Predicted score of edge $(u, i)$ produced by the model. |
| $\mathcal{E}^S$ | Current source edge set; in Problem (P) we assume $|\mathcal{E}^S| = N$. |
| $\mathcal{E}^T$ | Target edge set used to evaluate the performance drop in (P). |
| $N := |\mathcal{E}^S|$ | Number of edges in the source edge set. |
| $\mathbf{y}_S := (l(e))_{e \in \mathcal{E}^S}$ | Binary feedback vector of all source edges, $\mathbf{y}_S \in \{0,1\}^N$. |
| $\mathcal{E}^{S'} \subseteq \mathcal{E}^S$ | Selected subset of source edges; main decision variable in (P). |
| $\mathcal{E}_{\text{neg}}^{S'}$ | Selected negative edges: $\mathcal{E}_{\text{neg}}^{S'} = \{e \in \mathcal{E}^{S'} \mid l(e) = 0\}$. |
| $N_{\max}$ | Budget on the number of negative edges that are allowed to be kept, i.e., $|\mathcal{E}_{\text{neg}}^{S'}| \leq N_{\max}$. |
| $\mathcal{J}(\mathcal{E}^{S'})$ | Objective in (P): total performance drop on target-domain edges when using $(D_{S'}, l(\mathcal{E}^{S'}))$ instead of $(D_S, l(\mathcal{E}^S))$. |
| (P) | Original discrete combinatorial problem of selecting $\mathcal{E}^{S'}$ to maximize $\mathcal{J}(\mathcal{E}^{S'})$ under the budget constraint on negative edges. |
| (P)$_d$ | Decision version of (P): decide whether there exists $\mathcal{E}^{S'}$ with $\mathcal{J}(\mathcal{E}^{S'}) \geq B$ and $|\mathcal{E}_{\text{neg}}^{S'}| \leq N_{\max}$. |
| $[x]_+ := \max\{0, x\}$ | Positive-part operator used in the penalty formulation. |
| $\|\cdot\|_0$ | $\ell_0$ norm: counts the number of non-zero entries in a vector. |
| $\|\cdot\|_1$ | $\ell_1$ norm: sum of absolute values of vector entries. |
| $\lambda > 0$ | Penalty parameter controlling the strength of the negative-edge budget in the exact-penalty formulation. |
| $C$ | Constant offset used when shifting the objective by an exact-penalty transformation. |
| $(\bar{\text{P}})$ | Exact-penalty reformulation of (P) (Problem $(\bar{\text{P}})$), where the budget on $|\mathcal{E}_{\text{neg}}^{S'}|$ is enforced via a penalty term. |
| $g_\phi$ | GNN-based edge-weight (feedback) network used in the relaxed problem $(\text{P}^r)$. |
| $\phi$ | Parameters of the edge-weight network $g_\phi$. |
| $w_S := (l_1(e))_{e \in \mathcal{E}^S}$ | Continuous edge-weight vector; each weight is $l_1(e) = \sigma(g_\phi(e)) \in (0, 1)$. |
| $w_S'$ | Edge-weight vector restricted to $\mathcal{E}^{S'}$ (weights of selected edges). |
| $\mathbf{y}_S' := (l_2(e))_{e \in \mathcal{E}^{S'}}$ | Binary labels of edges in $\mathcal{E}^{S'}$ in the relaxed formulation (same labels as $\mathbf{y}_S$ but restricted). |
| $\odot$ | Hadamard (elementwise) product used in the $\ell_1$ penalty $\|w_S' \odot (1 - \mathbf{y}_S')\|_1$. |
| $L(\phi; \mathcal{E}^{S'})$ | Differentiable loss in the relaxed problem $(\text{P}^r)$, minimized with respect to $\phi$. |
| | Continued on next page |

| Symbol | Description |
|---|---|
| $\tau$ | Threshold on edge weights used to classify edges as causal vs. non-causal after solving $(\text{P}^r)$. |
| $(\text{P}^r)$ | Continuous relaxed problem where discrete selection in $(\text{P})$ is replaced by edge weights $w_S$ given by $g_\phi$. |

Table 5: Notation used in the NP-hardness proof and the reduction from Maximum Coverage.

| Symbol | Description |
|---|---|
| $(\text{P}_{\text{cov}})$ | Maximum Coverage optimization problem, whose decision version is NP-complete (hence $(\text{P}_{\text{cov}})$ is NP-hard). |
| $(\text{P}_{\text{cov}})_d$ | Decision version of Maximum Coverage: choose at most $k$ subsets to cover at least $B$ elements; this problem is NP-complete. |
| $\mathcal{G}$ | Ground set of elements in Maximum Coverage. |
| $\mathcal{S} = \{S_1, \ldots, S_m\}$ | Family of subsets of $\mathcal{G}$ in Maximum Coverage, where each $S_\ell \subseteq \mathcal{G}$. |
| $m := |\mathcal{S}|$ | Number of subsets in the Maximum Coverage instance. |
| $k$ | Budget on the number of subsets that can be chosen in Maximum Coverage; mapped to $N_{\max}$ in Problem $(\text{P})$. |
| $B$ | Coverage threshold in $(\text{P}_{\text{cov}})_d$ and also the objective threshold in $(\text{P})_d$. |
| $X \subseteq \mathcal{S}$ | Subfamily of subsets chosen in Maximum Coverage; corresponds to a selected subset $\mathcal{E}^{S'}$ in $(\text{P})$. |
| $t \in \mathcal{G}$ | A ground element; in the reduction each $t$ corresponds to one target-domain edge. |
| $e_t^T = (u^{(t)}, i^{(t)})$ | Target-domain edge constructed for each ground element $t \in \mathcal{G}$ in the reduction. |
| $\mathcal{E}^T = \{(u^{(t)}, i^{(t)}) : t \in \mathcal{G}\}$ | Constructed target edge set in the reduction, in one-to-one correspondence with $\mathcal{G}$. |
| $e_\ell \in \mathcal{E}^S$ | Source-domain edge corresponding to subset $S_\ell$ in the reduction. |
| $A_{\ell,t} \in \{0,1\}$ | Incidence indicator in the reduction: $A_{\ell,t} = 1$ iff element $t \in \mathcal{G}$ belongs to subset $S_\ell$. |
| OPT | Optimal objective value of an optimization problem instance. |
| $\leq_p$ | Polynomial-time many-one reduction between decision problems. |
| $\equiv_p$ | Polynomial-time equivalence between optimization problems. |
| $\leq_T$ | Polynomial-time Turing (oracle) reduction. |

# D  PROOF OF LEMMA 1

**Lemma 1.** *Let* $(\mathrm{P}_{\mathrm{cov}})$ *denote the Maximum Coverage problem. The decision version* $(\mathrm{P}_{\mathrm{cov}})_d$*, which is known to be NP-complete, reduces in polynomial time to the decision version of Problem (*P*), denoted as* $(\mathrm{P})_d$*:*

$$(\mathrm{P}_{\mathrm{cov}})_d \ \leq_p \ (\mathrm{P})_d.$$

*Furthermore, Problem (*$\overline{\mathrm{P}}$*) is polynomial-time equivalent to Problem (*P*)*

$$(\mathrm{P}) \ \equiv_p \ (\bar{\mathrm{P}}).$$

*Consequently, Problem (*$\overline{\mathrm{P}}$*) is NP-hard.*

*Proof.* Firstly, in the decision version of Maximum Coverage problem $(\mathrm{P}_{\mathrm{cov}})$, an instance of $(\mathrm{P}_{\mathrm{cov}})_d$ is given by a finite ground set $\mathcal{G}$, a family of subsets $\mathcal{S} = \{S_1, \ldots, S_m\} \subseteq 2^{\mathcal{G}}$, an integer budget $k \in \mathbb{N}$, and a threshold $B \in \mathbb{N}$. The decision problem $(\mathrm{P}_{\mathrm{cov}})_d$ asks whether there exists a subfamily $X \subseteq \mathcal{S}$ such that

$$|X| \leq k \quad \text{and} \quad \Big| \bigcup_{S \in X} S \Big| \geq B.$$

Equivalently, we want to know whether there is a choice of at most $k$ subsets that cover at least $B$ distinct elements of $\mathcal{G}$. It is well known that $(\mathrm{P}_{\mathrm{cov}})_d$ is NP-complete.

The decision version $(\mathrm{P})_d$ is obtained by adding a threshold $B$ on the objective function: given $B \in \mathbb{R}$, decide whether there exists $\mathcal{E}^{S'} \subseteq \mathcal{E}^S$ satisfying the constraint and $\mathcal{J}(\mathcal{E}^{S'}) \geq B$. We now construct, for any given instance $\langle \mathcal{G}, \mathcal{S}, k, B \rangle$ of $(\mathrm{P}_{\mathrm{cov}})_d$, an instance of $(\mathrm{P})_d$ in polynomial time such that the two instances are YES/NO-equivalent.

For each element $t \in \mathcal{G}$, we introduce a fresh target-domain edge $e_t^T := (u^{(t)}, i^{(t)})$, where the symbols $u^{(t)}$ and $i^{(t)}$ are just labels attached to $t$. We then define the target set

$$\mathcal{E}^T \ := \ \{(u^{(t)}, i^{(t)}) : t \in \mathcal{G}\}.$$

Thus there is a one-to-one correspondence between ground elements $t \in \mathcal{G}$ and target-domain edges $(u^{(t)}, i^{(t)}) \in \mathcal{E}^T$. For each subset $S_\ell \in \mathcal{S}$, we create one candidate source edge $e_\ell \in \mathcal{E}^S$. Thus we set

$$\mathcal{E}^S := \{e_1, \ldots, e_m\},$$

where $m = |\mathcal{S}|$. To encode the membership relation $t \in S_\ell$, we define the incidence indicator

$$A_{\ell,t} := \begin{cases} 1, & \text{if } t \in S_\ell, \\ 0, & \text{otherwise,} \end{cases} \qquad (\ell = 1, \ldots, m, \ t \in \mathcal{G}).$$

Note that each entry $A_{\ell,t}$ is a scalar in $\{0, 1\}$, not a matrix; for fixed $t$, the sum $\sum_\ell A_{\ell,t}$ simply counts how many selected subsets contain $t$. Recall that in Problem (P) the constraint involves the number of selected negative edges $|\mathcal{E}_{\mathrm{neg}}^{S'}|$. In our constructed instance, we simply label all candidate source edges as negative:

$$l(e_\ell) := 0 \quad \text{for all } \ell = 1, \ldots, m.$$

Thus $\mathbf{y}_S = (0, \ldots, 0) \in \{0, 1\}^m$, and for any choice $\mathcal{E}^{S'} \subseteq \mathcal{E}^S$, we have $\mathcal{E}_{\mathrm{neg}}^{S'} = \mathcal{E}^{S'}$, hence $|\mathcal{E}_{\mathrm{neg}}^{S'}| = |\mathcal{E}^{S'}|$. We set the budget parameter to match the Maximum Coverage budget:

$$N_{\max} := k.$$

Therefore the constraint $|\mathcal{E}_{\mathrm{neg}}^{S'}| \leq N_{\max}$ is equivalent to $|\mathcal{E}^{S'}| \leq k$, which directly parallels the constraint $|X| \leq k$ in Maximum Coverage. In Problem (P), the model parameters $\theta$ are fixed and only the subset $\mathcal{E}^{S'}$ is optimized. For the purpose of this hardness proof, it suffices to restrict attention to a subclass of instances where $\mathcal{F}_\theta^T$ is defined in an explicit way, which is clearly polynomial-time computable. For each target edge $(u^{(t)}, i^{(t)}) \in \mathcal{E}^T$ corresponding to $t \in \mathcal{G}$, we define

$$\mathcal{F}_\theta^T \big( u^{(t)}, i^{(t)} \mid \mathbb{D}_S, l(\mathcal{E}^S) \big) \ := \ 1,$$

and

$$\mathcal{F}_\theta^T\left(u^{(t)}, i^{(t)} \mid \mathbb{D}_{S'}, l(\mathcal{E}^{S'})\right) := 1 - \min\Big\{1, \sum_{e_\ell \in \mathcal{E}^{S'}} A_{\ell,t}\Big\}.$$

Intuitively, the value under $\mathbb{D}_S$ is a constant baseline 1, while the value under $\mathbb{D}_{S'}$ drops to 0 precisely when the element $t$ is "covered" by at least one selected source edge $e_\ell$, which is when $\sum_{e_\ell \in \mathcal{E}^{S'}} A_{\ell,t} \geq 1$. The quantity $\sum_{e_\ell \in \mathcal{E}^{S'}} A_{\ell,t}$ is an integer count, and thus $\min\{1, \sum_{e_\ell \in \mathcal{E}^{S'}} A_{\ell,t}\} \in \{0,1\}$ serves exactly as a binary indicator of whether $t$ is covered. Therefore the objective function in Problem (P) becomes

$$\begin{aligned}
\mathcal{J}(\mathcal{E}^{S'}) &= \sum_{(u,i) \in \mathcal{E}^T} \Big[ \mathcal{F}_\theta^T\left(u, i \mid \mathbb{D}_S, l(\mathcal{E}^S)\right) - \mathcal{F}_\theta^T\left(u, i \mid \mathbb{D}_{S'}, l(\mathcal{E}^{S'})\right) \Big] \\
&= \sum_{t \in \mathcal{G}} \min\Big\{1, \sum_{e_\ell \in \mathcal{E}^{S'}} A_{\ell,t}\Big\} \qquad (2) \\
&= \Big| \bigcup_{e_\ell \in \mathcal{E}^{S'}} S_\ell \Big|.
\end{aligned}$$

Therefore,

$$\exists \mathcal{E}^{S'} \subseteq \mathcal{E}^S : |\mathcal{E}_{\mathrm{neg}}^{S'}| \leq N_{\max}, \ \mathcal{J}(\mathcal{E}^{S'}) \geq B \quad \Longleftrightarrow \quad \exists X \subseteq \mathcal{S} : |X| \leq k, \ \Big| \bigcup_{S \in X} S \Big| \geq B,$$

where $X := \{S_\ell : e_\ell \in \mathcal{E}^{S'}\}$. The mapping from $\langle \mathcal{G}, \mathcal{S}, k, B \rangle$ to the corresponding instance of $(\mathrm{P})_d$ clearly runs in time polynomial in $|\mathcal{G}| + |\mathcal{S}|$. Thus, we have established a polynomial-time many-one reduction

$$(\mathrm{P}_{\mathrm{cov}})_d \ \leq_p \ (\mathrm{P})_d,$$

which shows the Problem $(\mathrm{P})_d$ is also NP-complete.

Then we can decide any instance of $(\mathrm{P})_d$ in polynomial time by a single call to this oracle: compute the optimal value, denoted as OPT and accept if and only if $\mathrm{OPT} \geq B$. This yields a polynomial-time Turing reduction

$$(\mathrm{P})_d \ \leq_T \ (\mathrm{P}).$$

Since $(\mathrm{P})_d$ is NP-complete, the optimization problem (P) is NP-hard.

Finally, by replacing the explicit $\ell_0$-budget constraint $|\mathcal{E}_{\mathrm{neg}}^{S'}| \leq N_{\max}$ by an exact penalty term of the form $\lambda \big[ |\mathcal{E}_{\mathrm{neg}}^{S'}| - N_{\max} \big]_+$ added to the objective, where $[x]_+ := \max\{0, x\}$ and $\lambda > 0$ is a penalty parameter. Choosing $\lambda$ larger than a known polynomial upper bound on the possible variation of $\mathcal{J}(\mathcal{E}^{S'})$ ensures that every optimal solution of the penalized problem is feasible for the original constrained problem and vice versa. Moreover, such a $\lambda$ can be chosen to be polynomially bounded and encoded using a number of bits polynomial in the input size, so the size of the instance remains polynomial. Therefore, the optimization problems (P) and $(\bar{\mathrm{P}})$ are polynomial-time equivalent,

$$(\mathrm{P}) \ \equiv_p \ (\bar{\mathrm{P}}).$$

Combining the above steps, we have

$$(\mathrm{P}_{\mathrm{cov}})_d \leq_p (\mathrm{P})_d, \quad (\mathrm{P})_d \in \mathrm{NP}, \quad (\mathrm{P})_d \leq_T (\mathrm{P}), \quad (\mathrm{P}) \equiv_p (\bar{\mathrm{P}}).$$

Since $(\mathrm{P}_{\mathrm{cov}})_d$ is NP-complete, Problem (P) is NP-hard, and by polynomial-time equivalence, Problem $(\bar{\mathrm{P}})$ is also NP-hard.

$\square$

**Lemma 2.** *Problem $(\overline{\mathrm{P}})$ is equivalent to problem (P), and problem $(\mathrm{P_r})$ provides relaxed solution:*

$$\mathcal{L}(\phi^*; \mathcal{E}^{S'}) - \lambda N_{max} \leq \overline{\mathcal{J}}((\mathcal{E}^{S'})^*)$$

*where $(\cdot)^*$ denotes the optimal solution, and $\lambda > 0$ is certain penalty parameter, $N \geq N_{max} \geq 0$.*

*Proof.* Since

$$\left[ C - \left( \mathcal{J}(\mathcal{E}^{S'}) - \lambda \left[ \|\mathbf{1} - \mathbf{y}_{S'}\|_0 - N_{\max} \right]_+ \right) \right] = \overline{\mathcal{J}}(\mathcal{E}^{S'}) ,$$

where $C$ is a constant, by the exact penalty function of the objective function with constraint in problem (P),

$$\begin{cases} \max \mathcal{J}(\mathcal{E}^{S'}) \\ \text{s.t. } N_{\max} \geq |\mathcal{E}_{\text{neg}}^{S'}| \end{cases} \Leftrightarrow \max \left( \mathcal{J}(\mathcal{E}^{S'}) - \lambda \left[ \|\mathbf{1} - \mathbf{y}_{S'}\|_0 - N_{\max} \right]_+ \Leftrightarrow \min \overline{\mathcal{J}}(\mathcal{E}^{S'})$$

So for certain $\lambda$, $(\mathrm{P}) \Leftrightarrow (\overline{\mathrm{P}})$. We also have:

$$\begin{aligned}
& \lambda \|(\mathbf{w}_{S'}) \odot (\mathbf{1} - \mathbf{y}_{S'})\|_1 \\
&= \lambda \sum_{e \in \mathcal{E}^{S'}} w_e (1 - y_e^{S'}) \leq \lambda \sum_{e \in \mathcal{E}^{S'}} (1 - y_e^{S'}) \\
&= \lambda \|\mathbf{1} - \mathbf{y}_{S'}\|_0 \\
&\leq \lambda N_{\max} + \lambda \left[ \|\mathbf{1} - \mathbf{y}_{S'}\|_0 - N_{\max} \right]_+,
\end{aligned} \tag{3}$$

then $\mathcal{L}(\phi^*; \mathcal{E}^{S'}) - \lambda N_{\max} \leq \overline{\mathcal{J}}((\mathcal{E}^{S'})^*)$. □

# E EXPERIMENTS SETTINGS AND EXTRA EXPERIMENTS

## E.1 DETAILS OF DATASETS

**Amazon Dataset.**[1] This extensive e-commerce collection encompasses item interactions across multiple categories. We select two domain pairs—Cloth & Sports, and Cloth & Video—for cross-domain recommendation experiments. The Cloth and Sports categories exhibit a moderate degree of relatedness, whereas Cloth and Video share relatively limited cross-domain knowledge.

**Douban Dataset.**[2] Originating from a popular music and movie online platform, this dataset supports two cross-domain tasks, with music and movie domains serving as target or source domains interchangeably.

In our experiments, we filter the dataset to keep users and items with at least 5 interactions and split the user history with the ratio of 8:1:1 for training, validation, and testing in the target domain for each user. The source domain is partitioned into training and validation sets with an 8:2 ratio. The detailed statistics of these datasets are provided in Appendix E.1.

Table 6: Dataset statistics. The subscript $o$ indicates *overlap*.

| Dataset | Domain | $|\mathcal{U}|$ | $|\mathcal{I}|$ | # Clicks | $|\mathcal{U}_o|$ | $|\mathcal{I}_o|$ |
|---|---|---|---|---|---|---|
| Amazon | Sports | 35,599 | 18,358 | 296,337 | 3,908 | 704 |
| | Cloth | 39,388 | 23,034 | 278,677 | | |
| | Video | 24,034 | 10,673 | 231,780 | 999 | 0 |
| | Cloth | 39,388 | 23,034 | 278,677 | | |
| Douban | Music | 16,041 | 40,405 | 1,140,090 | 14,000 | 0 |
| | Movie | 22,254 | 27,432 | 2,760,500 | | |

## E.2 DEFINITION OF EVALUATION METRICS

**Recall@K.** It is a metric that measures the fraction of relevant items retrieved out of all relevant items, which is formally defined as:

---

[1] http://jmcauley.ucsd.edu/data/amazon/index_2014.html
[2] https://recbole.s3-accelerate.amazonaws.com/CrossDomain/Douban.zip

$$\text{Recall@K} = \frac{1}{|\mathcal{U}|} \sum_{u \in \mathcal{U}} \frac{|\hat{R}(u) \cap R(u)|}{|R(u)|}, \tag{4}$$

where $\mathcal{U}$ is the set of all users, $\hat{R}(u)$ represents a ranked list of items that a model produces, and $R(u)$ represents a ground-truth set of items that the user has interacted with.

**NDCG@K.** It is a metric that measures ranking quality where positions are discounted logarithmically. It accounts for the position of the hits by assigning higher scores to hits at top ranks and it is formally defined as:

$$\text{NDCG@K} = \frac{1}{|\mathcal{U}|} \sum_{u \in \mathcal{U}} \left( \frac{1}{\sum_{i=1}^{\min(|R(u)|,K)} \frac{1}{\log_2(i+1)}} \quad \sum_{i=1}^{K} \frac{\delta(i \in R(u))}{\log_2(i+1)} \right). \tag{5}$$

### E.3 DETAILS OF THE BASELINES

The detailed introduction of the baselines compared with our model is as follows.

- **MF** Koren et al. (2009): This is a foundational collaborative filtering technique that models user-item interactions by decomposing the interaction matrix into low-dimensional latent factors. Specifically, it represents users and items as vectors in a shared latent space, capturing their characteristics through these latent factors.

- **LightGCN** He et al. (2020): LightGCN is a state-of-the-art single-domain recommender system designed for top-K recommendations. It leverages a simplified graph convolutional network (GCNs) to model collaborative signals by propagating user and item embeddings over a user-item interaction graph. Unlike traditional GCNs, LightGCN removes feature transformation and nonlinear activation, focusing solely on neighborhood aggregation to capture high-order connectivity. This streamlined approach enhances scalability and performance, making it a strong baseline for collaborative filtering tasks.

- **CMF** Singh & Gordon (2008b): CMF is a classical cross-domain recommender system that extends matrix factorization to multiple domains. It jointly factorizes the interaction matrices of both source and target domains, sharing latent factors for overlapping users or items. CMF adjusts prediction loss weights to balance contributions from each domain, enabling knowledge transfer while mitigating domain-specific noise.

- **EMCDR** Man et al. (2017): EMCDR introduces a cross-domain framework that aligns user embeddings between the source and target domains. It first trains separate embedding spaces for each domain using matrix factorization, then learns a mapping function to project source-domain user embeddings into the target-domain space. This mapping enables knowledge transfer for overlapping users while preserving domain-specific characteristics, making it effective for cross-domain recommendation tasks with shared users.

- **BiTGCF** Liu et al. (2020): BiTGCF is a cross-domain recommendation model that utilizes bi-directional graph convolutional networks to transfer knowledge between source and target domains. It learns robust user and item embeddings by modeling both shared preferences and domain-specific features, often using a triplet loss to refine the learned representations.

- **DTCDR** Zhu et al. (2019): DTCDR focuses on extracting domain-shared knowledge by integrating representations of overlapping users. It employs a deep neural network to model user-item interactions, capturing both domain-specific and shared features. By aligning representations of overlapping users across domains, DTCDR facilitates knowledge transfer while addressing negative transfer through careful feature disentanglement.

- **CAT-ART** Li et al. (2023): CAT-ART is a state-of-the-art cross-domain recommender that mitigates negative transfer by constructing a resilient global user representation. It employs an attention-driven transfer module to selectively transfer relevant information from the source domain to the target domain. The attention mechanism prioritizes domain-shared patterns, reducing the impact of irrelevant source-domain data.

- **UniCDR** Cao et al. (2023): UniCDR enhances cross-domain recommendation by facilitating the transfer of pertinent domain-shared information. It uses distinct user embeddings for each domain, augmented by interaction-level contrastive learning. This approach aligns user representations across domains while preserving domain-specific behaviors, improving recommendation accuracy.

- **CUT** Li et al. (2024): CUT is a recent cross-domain recommender that suppresses negative transfer by imposing explicit constraints based on target-domain user-similarity graphs. It constructs a graph of user similarities in the target domain and uses this structure to guide knowledge transfer from the source domain, ensuring only relevant information is incorporated. In our experiments, CUT is instantiated with both LightGCN and MF as backbone encoders, leveraging their strengths in modeling user-item interactions to enhance cross-domain performance.

For MF and LightGCN, we adopted the implementation from RecBole Zhao et al. (2021). For CMF Singh & Gordon (2008a), EMCDR Man et al. (2017), and DTCDR Zhu et al. (2019), we directly adopted the implementation from RecBole-CDR Zhao et al. (2022). For BiTGCF Liu et al. (2020), we made some modifications based on the RecBole-CDR Zhao et al. (2022) implementation; we tried changing the original cross entropy loss to BPR-loss Rendle et al. (2012) and found that it led to significant improvements across all datasets, so the results use the version with BPR-loss version. For CAT-ART Li et al. (2023) and UniCDR Cao et al. (2023), we integrated their original implementations into the RecBole-CDR Zhao et al. (2022) framework. For CUT Li et al. (2024), since the method in that paper is also based on RecBole-CDR Zhao et al. (2022), we adopted the implementation from the original paper.

### E.4 FURTHER ABLATION STUDY

To evaluate the efficacy of our proposed method in mitigating negative transfer, we benchmarked its performance on three representative backbones: BiTGCF, CUT, and LightGCN. We compared the performance of these models in three settings: (1) Single-domain setup, denoted as (**S**), (2) the Cross-domain setup, denoted as (**C**), and (3) the Cross-domain setup enhanced with our method, denoted as **+our method**. Fig 3 provides an intuitive visual comparison, while Table 7 presents a detailed breakdown of the results.

1. As illustrated in Fig 3, a key challenge becomes apparent when comparing the single-domain (S) and cross-domain (C) variants: cross-domain transfer does not always yield significant gains and can even be detrimental. For instance, as shown in Table 7, applying the cross-domain strategy to LightGCN on the Movie→Music dataset results in a substantial performance drop of **-9.07%** in Recall. This phenomenon of negative transfer is particularly consistent on the **Video→Cloth** dataset, where all three backbones suffer performance degradation; BiTGCF, for example, drops by a notable **-4.96%** in NDCG. Encouragingly, the integration of our proposed dataset regeneration method substantially ameliorates this problem. On this same challenging dataset, **+Our Method** not only reverses the negative trend but delivers remarkable gains, boosting the Recall of BiTGCF by **+21.17%** and CUT by **+21.52%** over their respective cross-domain counterparts.

2. When comparing the performance of LightGCN in single-domain versus cross-domain settings, it is often found that the single-domain version performs better. For example, on the **Movie→Music** dataset, the cross-domain model's Recall@10 drops to 0.0972 from the single-domain's 0.1069, a significant degradation of **9.07%**. This indicates that simply merging data from two distinct domains can cause severe negative transfer due to their inherent discrepancies. However, this phenomenon is effectively mitigated by more advanced cross-domain recommendation methods. On that same **Movie→Music** dataset, dedicated CDR models like **BiTGCF** and **CUT** achieve much higher Recall@10 scores of **0.1209** and **0.1205** respectively, outperforming the single-domain baseline by an impressive **13.10%** and **12.72%**. This pattern is even more pronounced on the **Music→Movie** dataset, where **CUT** reaches a remarkable **0.1393** in Recall—a massive **35.11%** improvement over the single-domain performance—demonstrating the clear superiority of sophisticated cross-domain architectures in overcoming negative transfer.

3. After applying **+Our Method**, this trend is dramatically reversed. The Recall for CUT improves by a remarkable **+21.52%**, and for BiTGCF by **+21.17%**. This success stems from our model's ability to effectively identify and prune these noisy or irrelevant interactions, thereby preventing them from negatively impacting the model's training and allowing the genuine transfer of useful knowledge.

### E.5 CASE STUDY

Figure 4 illustrates two overlap users with very different cross-domain preferences. Anonymous user A consistently interacts with music-related films in the movie domain (e.g., Begin Again, La La Land, Whiplash) and with jazz albums

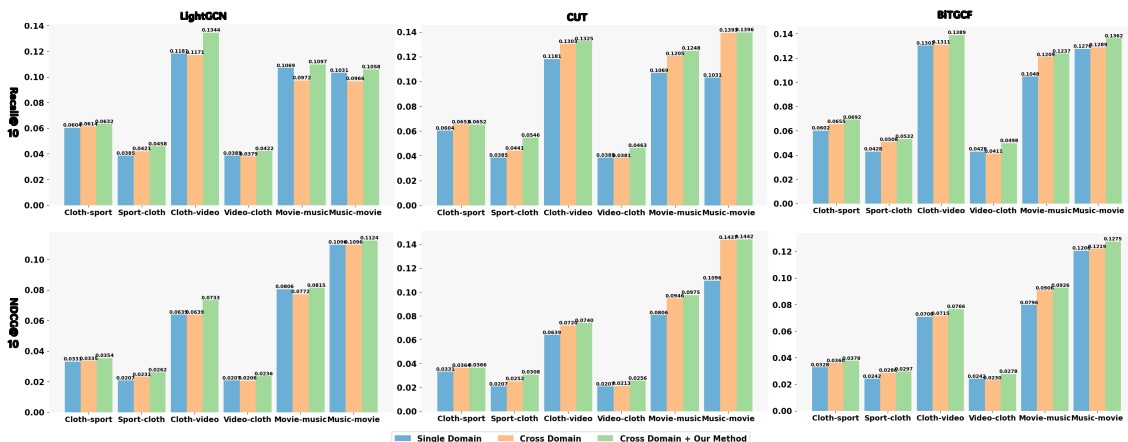

Figure 3: Ablation study on negative transfer

Table 7: Ablation study on different backbones. (S) denotes the single-domain version, while (C) represents the cross-domain version. +Our Method indicates the addition of our proposed modules to (C). The percentage change for (C) is relative to (S), and for +Our Method, it is relative to (C). Improvements are in orange, drops in cyan, with color intensity scaling with the magnitude of change.

| Backbone | Model | Amazon | | | | | | | | Douban | | | |
|---|---|---|---|---|---|---|---|---|---|---|---|---|---|
| | | Cloth→Sport | | Sport→Cloth | | Cloth→Video | | Video→Cloth | | Movie→Music | | Music→Movie | |
| | | R | N | R | N | R | N | R | N | R | N | R | N |
| LightGCN | (S) | 0.0604 | 0.0331 | 0.0385 | 0.0207 | 0.1181 | 0.0639 | 0.0385 | 0.0207 | 0.1069 | 0.0806 | 0.1031 | 0.1096 |
| | (C) | 0.0614 +1.66% | 0.0335 +1.21% | 0.0421 +9.35% | 0.0231 +11.59% | 0.1171 -0.85% | 0.0639 +0.00% | 0.0379 -1.56% | 0.0206 -0.48% | 0.0972 -9.07% | 0.0772 -4.22% | 0.0966 -6.30% | 0.1096 +0.00% |
| | +Our Method | 0.0632 +2.93% | 0.0354 +5.67% | 0.0458 +8.79% | 0.0262 +13.42% | 0.1344 +14.77% | 0.0733 +14.71% | 0.0422 +11.35% | 0.0236 +14.56% | 0.1097 +12.86% | 0.0815 +5.57% | 0.1058 +9.52% | 0.1124 +2.55% |
| CUT | (S) | 0.0604 | 0.0331 | 0.0385 | 0.0207 | 0.1181 | 0.0639 | 0.0385 | 0.0207 | 0.1069 | 0.0806 | 0.1031 | 0.1096 |
| | (C) | 0.0653 +8.11% | 0.0364 +9.97% | 0.0441 +14.55% | 0.0252 +21.74% | 0.1303 +10.33% | 0.0720 +12.68% | 0.0381 -1.04% | 0.0213 +2.90% | 0.1205 +12.72% | 0.0946 +17.37% | 0.1393 +35.11% | 0.1437 +31.11% |
| | +Our Method | 0.0652 -0.15% | 0.0366 +0.55% | 0.0546 +23.81% | 0.0308 +22.22% | 0.1325 +1.69% | 0.0740 +2.78% | 0.0463 +21.52% | 0.0256 +20.19% | 0.1248 +3.57% | 0.0975 +3.07% | 0.1396 +0.22% | 0.1442 +0.35% |
| BiTGCF | (S) | 0.0602 | 0.0328 | 0.0428 | 0.0242 | 0.1302 | 0.0708 | 0.0428 | 0.0242 | 0.1048 | 0.0796 | 0.1276 | 0.1206 |
| | (C) | 0.0655 +8.80% | 0.0360 +9.76% | 0.0508 +18.69% | 0.0286 +18.18% | 0.1311 +0.69% | 0.0715 +0.99% | 0.0411 -3.97% | 0.0230 -4.96% | 0.1209 +15.36% | 0.0906 +13.82% | 0.1289 +1.02% | 0.1219 +1.08% |
| | +Our Method | 0.0692 +5.65% | 0.0378 +5.00% | 0.0532 +4.72% | 0.0297 +3.85% | 0.1389 +5.95% | 0.0766 +7.13% | 0.0498 +21.17% | 0.0278 +20.87% | 0.1237 +2.32% | 0.0926 +2.21% | 0.1362 +5.66% | 0.1275 +4.59% |

and soundtracks in the music domain (e.g., Kind of Blue, Blue Train, Whiplash OST, La La Land OST). For this user, edges between "music films" and "jazz music" are truly informative and should be preserved. Anonymous user B, in contrast, mainly watches hot-blooded competition films such as Rocky, Rocky Balboa, Creed, and Million Dollar Baby, and listens to energetic rock anthems such as Chasing the Dreamers and We Are Mayday; B happens to watch Whiplash once because it is also hot-blooded films. A standard CDR model (BiTGCF) treats the interaction with Whiplash as strong evidence of a jazz preference and therefore assigns a high similarity between user B and the jazz group (Cos = 0.4095, rank = 8), producing a false positive recommendation of jazz soundtracks. Our Gen/Del framework instead filters the non-causal movie interaction (Whiplash for user B) and re-optimizes the data, which substantially weakens the spurious link from B to jazz (Cos = 0.2406, rank = 142) while keeping B's connections to rock music. In this way, the same movie (Whiplash) is treated as causal for user A but spurious for user B, and Gen/Del correctly steers recommendations towards jazz for A and towards motivational rock tracks for B, thereby mitigating popularity-driven false positives.

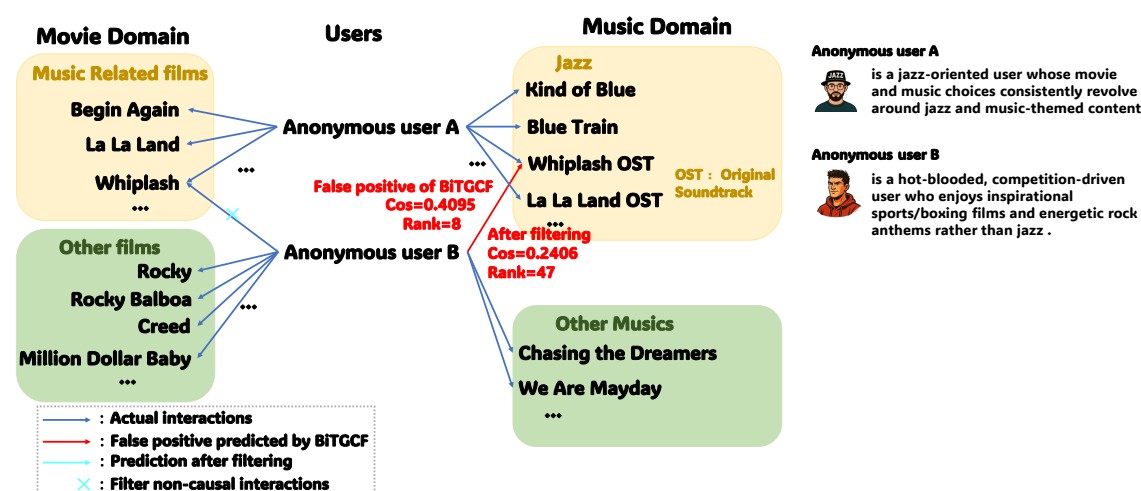

Figure 4: A case study for filtering process.

## F    COMPUTATION COMPLEXITY ANALYSIS

In this section, we provide a detailed analysis of the time and space complexity of the proposed dataset regeneration framework, based on Algorithm 1. The number of non-overlapping target users is $|\mathcal{U}^{T'}| = |\mathcal{U}^T| - |\mathcal{U}^O|$. In many real-world scenarios, such as the Amazon datasets used in our experiments, overlap is sparse (e.g., $|\mathcal{U}^O|/|\mathcal{U}^T| < 10\%$), leading to $|\mathcal{U}^{T'}| \approx |\mathcal{U}^T|$; in datasets with high overlap like Douban (e.g., up to 87%), $|\mathcal{U}^{T'}|$ is significantly smaller, which takes less computational cost. The surrogate model $\mathcal{F}_\theta$ and edge-weighting model $g_\phi$ are implemented as lightweight GNNs (e.g., LightGCN He et al. (2020) with $L$ layers and embedding dimension $d$), where a single forward pass has time complexity $O((|\mathcal{E}| + |\mathcal{V}|) \cdot L \cdot d) \approx O(|\mathcal{E}| \cdot d)$ for a graph with $|\mathcal{E}|$ edges and $|\mathcal{V}|$ nodes, assuming sparse matrix operations. The framework consists of three phases:

**Pre-training Phase (Part 1):** This involves $E_{pre}$ epochs of training $\mathcal{F}_\theta$ on the combined graph with $|\mathcal{U}^S| + |\mathcal{U}^T| - |\mathcal{U}^O|$ users, $|\mathcal{I}^S| + |\mathcal{I}^T|$ items, and $|\mathcal{E}^S| + |\mathcal{E}^T|$ edges. Each epoch requires a forward pass, BPR loss computation over sampled pairs (typically $O(|\mathcal{E}^S| + |\mathcal{E}^T|)$ operations), and backpropagation. Masking affects a small subset of edges ($O(|\mathcal{U}^O|)$ on average), adding negligible overhead. Total time complexity: $O(E_{pre} \cdot (|\mathcal{E}^S| + |\mathcal{E}^T|) \cdot d)$. Space complexity is dominated by storing the graph structure ($O(|\mathcal{E}^S| + |\mathcal{E}^T|)$) and embeddings ($O((|\mathcal{U}^S| + |\mathcal{U}^T| + |\mathcal{I}^S| + |\mathcal{I}^T|) \cdot d)$), with temporary buffers for masked edges adding $O(|\mathcal{U}^O|)$ space.

**Generation Phase (Part 2):** For each of the $|\mathcal{U}^{T'}|$ non-overlapping users, we compute scores $\hat{y}_{u,i^S}$ for all $|\mathcal{I}^S|$ source items using the frozen $\mathcal{F}_\theta$. A single forward pass on the combined graph yields all user and item embeddings ($O((|\mathcal{E}^S| + |\mathcal{E}^T|) \cdot d)$ time), after which scores are computed via inner products in $O(|\mathcal{U}^{T'}| \cdot |\mathcal{I}^S| \cdot d)$ time. Selecting top-$k$ per user adds $O(|\mathcal{U}^{T'}| \cdot |\mathcal{I}^S| \cdot \log k)$ via sorting or heaps (negligible for small $k$). The augmented edge set $|\mathcal{E}^S_{aug}| = |\mathcal{E}^S| + k \cdot |\mathcal{U}^{T'}|$. Total time: $O((|\mathcal{E}^S| + |\mathcal{E}^T| + |\mathcal{U}^{T'}| \cdot |\mathcal{I}^S|) \cdot d)$. This phase scales with $|\mathcal{U}^{T'}|$, which is lower in high-overlap datasets like Douban. Space complexity: $O((|\mathcal{U}^S| + |\mathcal{U}^T| + |\mathcal{I}^S| + |\mathcal{I}^T|) \cdot d + |\mathcal{E}^S| + |\mathcal{E}^T| + k \cdot |\mathcal{U}^{T'}|)$ for embeddings and augmented edges.

**Counterfactual Filtering Phase (Part 3):** Over $E_{filter}$ epochs, we train $g_\phi$ on the augmented source graph $\mathbb{D}^{aug}_S$ with $|\mathcal{E}^S_{aug}|$ edges. Each epoch involves: (i) a forward pass of $g_\phi$ to compute edge weights ($O(|\mathcal{E}^S_{aug}| \cdot d)$ time); (ii) a weighted forward pass of the frozen $\mathcal{F}_\theta$ on the combined graph ($O((|\mathcal{E}^S_{aug}| + |\mathcal{E}^T|) \cdot d)$ time); and (iii) loss computation over $|\mathcal{E}^T|$ target interactions plus regularization ($O(|\mathcal{E}^T| + |\mathcal{E}^S_{aug}|)$ time). Backpropagation scales similarly. Total time:

$O(E_{filter} \cdot (|\mathcal{E}^S_{aug}| + |\mathcal{E}^T|) \cdot d) \approx O(E_{filter} \cdot (|\mathcal{E}^S| + |\mathcal{E}^T| + k \cdot |\mathcal{U}^{T'}|) \cdot d)$. Space complexity: $O((|\mathcal{U}^S| + |\mathcal{U}^T| + |\mathcal{I}^S| + |\mathcal{I}^T|) \cdot d + |\mathcal{E}^S_{aug}| + |\mathcal{E}^T|)$ for embeddings, augmented graph, and edge weights $\mathbf{w}_S \in (0,1)^{|\mathcal{E}^S_{aug}|}$.

Overall time complexity: $O((E_{pre} + E_{filter}) \cdot (|\mathcal{E}^S| + |\mathcal{E}^T|) \cdot d + |\mathcal{U}^{T'}| \cdot |\mathcal{I}^S| \cdot d)$. Overall space complexity: $O((|\mathcal{U}^S| + |\mathcal{U}^T| + |\mathcal{I}^S| + |\mathcal{I}^T|) \cdot d + |\mathcal{E}^S| + |\mathcal{E}^T| + k \cdot |\mathcal{U}^{T'}|)$.

**Algorithm 1** Generate-and-Filter Dataset Regeneration

**Input:** Source domain data $\mathbb{D}_S = (\mathcal{U}^S, \mathcal{I}^S, \mathcal{E}^S)$;
Target domain data $\mathbb{D}_T = (\mathcal{U}^T, \mathcal{I}^T, \mathcal{E}^T)$;
Non-overlapping target users $\mathcal{U}^{T'} = \mathcal{U}^T \setminus \mathcal{U}^O$;
Pre-train epochs $E_{pre}$; Filtering epochs $E_{filter}$;
Generation top-$k$ $k$; Pruning threshold $\tau$; Regularizer $\lambda$
**Output:** Regenerated source dataset $\mathbb{D}_{S'} = (\mathcal{U}^S \cup \mathcal{U}^T, \mathcal{I}^S, \mathcal{E}')$

**Part 1: Pre-train Surrogate Model (Sec 2.2)**
  Initialize surrogate model $\mathcal{F}_\theta$ with parameters $\theta$
  **for** *epoch = 1* **to** $E_{pre}$ **do**
       Select subset of overlapping users $\mathcal{U}_{mask} \subset \mathcal{U}^O$
          Get masked edges $\mathcal{E}_{mask}^S$ for users in $\mathcal{U}_{mask}$
          ▷ Optimize via BPR loss (Eq. 1) on two tasks
       Update $\theta$ by minimizing $\mathcal{L}_{pre}$ (whole-graph + masked-edge reconstruction)
  **end**
**Freeze** parameters $\theta$ of the surrogate model $\mathcal{F}_\theta$

**Part 2: Generation Phase (Sec 2.2)**
    $\mathcal{E}_{\text{gen}} \leftarrow \emptyset$ ▷ Initialize synthetic (generated) edge set
  **for** *each non-overlapping user $u \in \mathcal{U}^{T'}$* **do**
       ▷ Use the frozen $\mathcal{F}_\theta$ to get scores
       Calculate scores $\hat{y}_{u,i^S} \leftarrow \mathcal{F}_\theta(u, i^S)$ for all $i^S \in \mathcal{I}^S$
       $\mathcal{I}_{top-k}^S \leftarrow$ set of top-$k$ items for user $u$ based on scores $\hat{y}_{u,i^S}$
       $\mathcal{E}_{\text{gen}} \leftarrow \mathcal{E}_{\text{gen}} \cup \{(u, i^S) \mid i^S \in \mathcal{I}_{top-k}^S\}$
  **end**
  $\mathcal{E}_{aug}^S \leftarrow \mathcal{E}^S \cup \mathcal{E}_{\text{gen}}$ ▷ Augmented source edge set
  Let $\mathbb{D}_S^{aug} = (\mathcal{U}^S \cup \mathcal{U}^T, \mathcal{I}^S, \mathcal{E}_{aug}^S)$

**Part 3: Counterfactual Interaction Filtering (Sec 2.3)**
  Initialize edge-weighting GNN $g_\phi$ with parameters $\phi$
  **for** *epoch = 1* **to** $E_{filter}$ **do**
       ▷ Get continuous weights for all augmented edges
       $\mathbf{w}_S \leftarrow \sigma(g_\phi(\mathbb{D}_S^{aug}))$ ▷ $\mathbf{w}_S \in (0,1)^{|\mathcal{E}_{aug}^S|}$
       ▷ Update $\phi$ to minimize target performance of the **frozen** $\mathcal{F}_\theta^T$
       ▷ This is the inverse optimization problem (Eq. $P_r$)
       $\mathcal{L}_{filter} \leftarrow \sum_{(u,i) \in \mathcal{E}_{\text{pos}}^T} \text{PredLoss}(\mathcal{F}_\theta^T(u, i \mid \mathbb{D}_S^{aug}, \mathbf{w}_S)) + \lambda \cdot \text{Reg}(\mathbf{w}_S)$
          Update $\phi$ using gradient $\nabla_\phi \mathcal{L}_{filter}$ ▷ $\theta$ remains frozen
  **end**
  Get final converged edge weights $\mathbf{w}_S^* \leftarrow \sigma(g_\phi(\mathbb{D}_S^{aug}))$
    ▷ Filter edges based on threshold $\tau$ (Sec 2.3, line 250)
  $\mathcal{E}' \leftarrow \{e \in \mathcal{E}_{aug}^S \mid w_e^* < \tau\}$
  $\mathbb{D}_{S'} \leftarrow (\mathcal{U}^S \cup \mathcal{U}^T, \mathcal{I}^S, \mathcal{E}')$

**return** $\mathbb{D}_{S'}$

## F.1 THE USE OF LARGE LANGUAGE MODELS

In the preparation of this manuscript, we utilized Large LLMs to assist with several tasks. Primarily, LLMs were employed to help format and input complex experimental results into LaTeX tables. Furthermore, they served as a tool for proofreading the manuscript to identify potential logical inconsistencies and correct English grammatical errors. Finally, LLMs were also used to assist in translating some of the content into English.

