# OpenReview forum: "Dataset Regeneration for Cross Domain Recommendation"
_ICLR.cc/2026/Conference — Submitted to ICLR 2026_

### Official Review · Reviewer_MGmq · 2025-10-23

**Soundness:** 3
**Presentation:** 3
**Contribution:** 3
**Rating:** 8
**Confidence:** 3

**Summary:**

The paper proposes a data-centric framework, Generate-and-Filter (Gen/Del), for
cross-domain recommendation (CDR). Instead of focusing on model-level transfer, the
authors address data sparsity and negative transfer by regenerating a causal and
denoised source-domain dataset. The framework consists of two stages:
(1) Generation phase: A self-supervised model generates synthetic source-domain
interactions for users who exist only in the target domain, using masked-edge
reconstruction and BPR loss.
(2) Filtering phase: A counterfactual inference module assigns causal importance
weights to each generated or existing edge and filters out non-causal or spurious ones.
The resulting regenerated dataset can be plugged into any backbone recommender
(e.g., LightGCN, CUT, BiTGCF). Experiments on Douban and Amazon datasets show
consistent improvements across multiple backbones, with gains up to 23.8% in
Recall@10.

**Strengths:**

•	Originality: Presents a fresh, data-centric perspective on CDR, shifting focus from model-level transfer to dataset regeneration. The integration of causal counterfactual filtering with GNN-based representation learning is particularly innovative.
	•	Quality: Methodology is sound and well-formulated, with strong empirical results across multiple datasets and backbone models. Ablation studies effectively demonstrate the framework’s ability to mitigate negative transfer.
	•	Clarity: The paper is clearly written and well-structured, with intuitive explanations and informative figures.
	•	Significance: The framework is model-agnostic and has broad applicability, offering a principled foundation for future research on causal data manipulation and transfer learning.

Overall, the work is conceptually original, empirically convincing, and highly relevant to data-centric and causal learning in recommender systems.

**Weaknesses:**

(1) Limited Analysis of Computational Cost and Scalability
While the proposed Generate-and-Filter framework is conceptually appealing, the paper lacks a systematic evaluation of its computational overhead. The counterfactual filtering stage requires training an additional GNN and repeatedly assessing target-domain performance, which could be computationally intensive for large-scale datasets. However, the paper provides no quantitative analysis of runtime, memory consumption, or scaling behavior with respect to dataset size, leaving the practicality of the approach for industrial-scale recommender systems uncertain.

(2) Incomplete Symbol Definitions in the Counterfactual Interaction Filtering Section
Several key symbols in Section 2.3—such as F_t^s, y_i, E_t^s, and the mapping l(E_t)—are introduced without explicit definitions or consistent explanations. This lack of clarity makes the mathematical formulation difficult to follow and reproduce. A concise summary table of notations or explicit variable definitions would greatly enhance readability and reproducibility.

(3) Lack of Qualitative Analysis and Interpretability of Filtering Results
Although the paper presents quantitative improvements in metrics such as Recall@10 and NDCG@10, it lacks qualitative analysis of the filtering process. There are no examples or visualizations illustrating which user–item edges are pruned or retained by the counterfactual filtering stage. Without such interpretability analysis, it is difficult to understand what types of interactions the model identifies as causal versus spurious.

(4) Unclear Contribution of the Generation Phase
Ablation results suggest that most of the performance gains arise from the counterfactual filtering module rather than the data generation phase. However, the paper does not analyze the characteristics or quality of the generated interactions—such as their distribution, overlap with observed data, or effect on coverage. Consequently, the empirical contribution and necessity of the generation component remain ambiguous.

**Questions:**

(1) Address scalability and efficiency concerns
Providing details on the model’s runtime, computational cost, and resource usage
during experiments would help readers better understand the practical feasibility and
efficiency of the proposed framework.
(2) Deeper analysis of generation and filtering behavior
Analyzing how the generation phase adds synthetic edges and how the
counterfactual filtering module removes or retains interactions in practice would help
readers better understand the model’s decision behavior and its contribution to
performance improvements.

---

> ### Author Response · Authors · 2025-11-19
> **Response to reviewer's Concern about scalibility and problems of the symbols**
>
> **Comment:**
> >Limited Analysis of Computational Cost and Scalability While the proposed Generate-and-Filter framework is conceptually appealing, the paper lacks a systematic evaluation of its computational overhead. The counterfactual filtering stage requires training an additional GNN and repeatedly assessing target-domain performance, which could be computationally intensive for large-scale datasets. However, the paper provides no quantitative analysis of runtime, memory consumption, or scaling behavior with respect to dataset size, leaving the practicality of the approach for industrial-scale recommender systems uncertain.
>
> **Response:** We thank the reviewer for the comment. We clarify that the practicality of our framework is a key design consideration. To provide a systematic answer, **we have added a Computation Complexity Analysis section in Appendix F.**
>
>
> **Comment:**
> >Incomplete Symbol Definitions in the Counterfactual Interaction Filtering Section Several key symbols in Section 2.3—such as F_t^s, y_i, E_t^s, and the mapping l(E_t)—are introduced without explicit definitions or consistent explanations. This lack of clarity makes the mathematical formulation difficult to follow and reproduce. A concise summary table of notations or explicit variable definitions would greatly enhance readability and reproducibility.
>
> **Response:** Thank you for this helpful comment and for carefully checking the mathematical formulation. We re-examined Section 2.3 in the submitted version and did not find the exact symbols you mentioned (such as $F_t^s$, $y_i$, $E_t^s$, or $l(E_t)$); we suspect these may have arisen from a misreading of symbols like $\mathcal{F}^T_\theta$, $y_{ui}$, $\mathcal{E}^S/\mathcal{E}^T$, and $l(\mathcal{E}^S)$, or from their definitions being too scattered and thus hard to follow. That said, we fully agree with your underlying point that the notation in the counterfactual interaction filtering section should be self-contained and explicit.
>
> In the revised manuscript, we have addressed this concern in two ways. First, we added a dedicated "Notation" section with concise tables summarizing all symbols used in the appendix, including the prediction function $\mathcal{F}^T_\theta$, feedback labels $y_{ui}$, source/target edge sets $\mathcal{E}^S, \mathcal{E}^T$, and the feedback mapping $l(\cdot)$. Second, we moved the preliminaries from the appendix to the main text and enriched them with more intuitive explanations, so that the counterfactual filtering procedure can be understood and reproduced without referring back and forth. We believe these changes substantially improve the clarity and reproducibility of the mathematical formulation.

---

> ### Author Response · Authors · 2025-11-19
> **Response to Reviewer's Concern about Interpretability of Filtering Process andContribution of Generation Process**
>
> **Comment:**
> >Lack of Qualitative Analysis and Interpretability of Filtering Results Although the paper presents quantitative improvements in metrics such as Recall@10 and NDCG@10, it lacks qualitative analysis of the filtering process. There are no examples or visualizations illustrating which user–item edges are pruned or retained by the counterfactual filtering stage. Without such interpretability analysis, it is difficult to understand what types of interactions the model identifies as causal versus spurious.
>
> Response: Thanks for the comment. **We have provided an intuitive case study in Appendix E.** Here we provide a summary of it.  As illustrated in Figure 4 in the paper, we present a case study comparing two users who watched the same movie (Whiplash) but with different underlying intents. While the baseline BiTGCF treats this interaction uniformly, leading to false-positive jazz recommendations for User B (who prefers rock/sports), our Gen/Del framework correctly identifies this interaction as non-causal for User B. By filtering this spurious edge, our method effectively disentangles user interest, steering recommendations towards their true preference (Rock) while preserving the valid connection for User A (Jazz).
>
>
> **Comment:**
>
> >Unclear Contribution of the Generation Phase Ablation results suggest that most of the performance gains arise from the counterfactual filtering module rather than the data generation phase.
>
> **Response:**
>
> We thank the reviewer for the observation regarding the ablation results. We acknowledge that the **standalone performance of the generation module (+Gen) can be limited by the introduction of noise**. However, we respectfully argue that **the generation phase is necessary for the whole process**. This is evidenced by the substantial **relative performance gains** across multiple datasets.
>
>
> We list some examples in Table 2 to show the additional improvement brought by generation is massive:
>
>
> | Datasets | Metric      | +Del alone | +Gen/Del| Additional abs. | **Relative gain over +Del** |
> |-------------------------|------------------|----------------|---------------------|-------------------|-------------------------------------|
> | Video → Cloth         | Recall@10  | +12.41%   | +21.17%        | +8.76%         | **+70.6%**                       |
> | Video → Cloth         | NDCG@10    | +11.30%   | +20.87%        | +9.57%         | **+84.7%**                       |
> | Cloth → Sport         | Recall@10  | +2.60%    | +5.65%         | +3.05%         | **+117.3%**                      |
> | Movie → Music         | Recall@10  | +0.58%    | +2.32%         | +1.74%         | **+300.0%**                      |
>
> **Comment:**
>
> >The paper does not analyze the characteristics or quality of the generated interactions—such as their distribution, overlap with observed data, or effect on coverage. Consequently, the empirical contribution and necessity of the generation component remain ambiguous.
>
>
> **Response:**
>
>
> Thanks for the comments. We wish to clarify that the ambiguity of the Generation phase's contribution.
>
>
> We chose a uniform top-k approach to function as a "high-recall candidate pool generator."
>
> The generation phase is **designed to be loose to create a sufficiently large pool of candidate bridges**. Its primary goal is to maximally alleviate sparse overlap by ensuring every non-overlapping target user gains potential connectivity to the source domain. We **accept the introduction of noise/false positives at this stage because the subsequent counterfactual filtering stage is explicitly designed to handle these noises**. It acts as a gatekeeper, pruning not only original spurious interactions but also the low-quality edges introduced during generation.

---

> ### Author Response · Authors · 2025-11-26
> **Gentle Reminder**
>
> Dear Reviewer MGmq,
>
> As the discussion period is approaching its end, we wanted to kindly check if our previous response clarified your questions. We are happy to provide any further details if needed.
>
> Thank you for your time and effort in reviewing our paper.
>
> Best regards, Authors

---

### Official Review · Reviewer_eJuQ · 2025-10-31

**Soundness:** 3
**Presentation:** 3
**Contribution:** 2
**Rating:** 4
**Confidence:** 3

**Summary:**

This paper focuses on the cross-domain recommendation (CDR) task and addresses key challenges, including sparse user overlap across domains and negative transfer caused by spurious correlations in heterogeneous data. To tackle these issues, the authors propose a dataset regeneration framework that (1) generates high-confidence candidate interactions to link non-overlapping users and items, and (2) applies a causal-inference-inspired filtering process to remove spurious interactions from both the generated and original data. This approach enhances the causal connection between source and target domains. When integrated with recommendation models such as LightGCN, BiTGCF, and CUT, it substantially improves target-domain performance, achieving up to 23.81% gain in Recall@10 and 22.22% in NDCG@10.

**Strengths:**

1. This paper focuses on the cross-domain recommendation (CDR) task and addresses two major challenges: sparse user overlap across domains and negative transfer caused by spurious correlations in heterogeneous data.
2. To tackle these challenges, the authors propose a dataset regeneration framework. This approach strengthens the causal connection between the source and target domains.
3. The proposed framework, when integrated with recommendation models such as LightGCN, BiTGCF, and CUT, substantially improves target-domain performance.

**Weaknesses:**

1. The core argument of this paper is that prior work primarily addresses sparse overlap and negative transfer at the model level, whereas this work tackles these challenges from a data-centric perspective. In fact, in the cross-domain recommendation (CDR) field, several studies have already explored data-centric solutions, such as [1][2][3]. The authors also provide a comparative analysis between their approach and these existing data-centric methods.

[1]https://arxiv.org/pdf/2405.20710
[2]https://arxiv.org/abs/2307.13910
[3]https://dl.acm.org/doi/10.1145/3626772.3657902

2. There is an inconsistency between the paper title in the main text and the title on OpenReview. The authors should ensure that the titles are consistent before submission.

3. The proposed framework is divided into two stages: generation followed by filtering.
- For the generation stage, the authors employ self-supervised pretraining, which is a common practice in graph learning, and therefore this stage lacks significant novelty.
- For the filtering stage, the authors adopt counterfactual interaction filtering. It would be helpful to clarify the motivation for using this technique compared with existing filter-based methods. Are there unique challenges that the counterfactual approach specifically addresses?

**Questions:**

see weaknesses

---

> ### Author Response · Authors · 2025-11-19
> **Clarification of the data-centric perspective and revision of inconsistency of the paper title**
>
> **Comment:**
>
> > The core argument of this paper is that prior work primarily addresses sparse overlap and negative transfer at the model level, whereas this work tackles these challenges from a data-centric perspective. In fact, in the cross-domain recommendation (CDR) field, several studies have already explored data-centric solutions, such as [1][2][3]. The authors also provide a comparative analysis between their approach and these existing data-centric methods.
>
>
>
> **Response:**
>
> We thank the reviewer for highlighting prior data-centric works in CDR [1–3]. While these methods indeed incorporate data-centric ideas, they fundamentally operate as **data augmentation**, whereas our method performs **dataset regeneration**. This distinction is substantive and leads to different capabilities and goals. Below, we clarify the differences.
>
> **1. Sample/Representation-level vs. Dataset-level:**
>
> Prior works [1–3] perform data augmentation inside model training:
>
> - i²VAE[1] generates pseudo-sequences via a VAE during training.
>
> - DIDA-CDR [2] interpolates representations in the model’s latent space.
>
> - CrossAug [3] conducts cross-reconstruction and alignment within the encoder.
>
> In all cases, augmentation is local, executed within each batch while the model is still being optimized.
> Thus, the model must simultaneously:
>
> - Learn its parameters.
> - Determine what augmented data is beneficial.
>
> Because the model representation is continually changing during training, it cannot support global, stable causal evaluation of which source-domain interactions truly benefit the target domain.
>
> In contrast, our method introduces a two-stage, offline dataset-level regeneration process:
>
> - **Surrogate pretraining:** We pretrain a prediction model $\mathcal{F}_{\theta}^{T}$ until convergence, obtaining a stable and reliable oracle.
> - **Counterfactual dataset intervention:** With the surrogate frozen, we evaluate the causal effect of each source-domain interaction on target-domain performance.
>
>
> This enables a **global, dataset-wide identification of harmful interactions** and regeneration of a modified dataset before any CDR backbone is trained. This offline counterfactual filtering addresses “root-level” data issues (e.g., interactions that are valid within the source domain but causally detrimental to the target domain), which cannot be detected by in-training augmentation methods that only enforce local consistency.
>
>
> **2. Model-specific vs. Model-agnostic:**
>
> As augmentations in [1,2,3] are implemented within each model’s architecture (e.g., VAE-based sequence generation [1], representation interpolation [2], encoder-based cross-reconstruction [3]), they inherently remain **model-specific**.
>
>
> Our method **optimizes the data, not the model**:
> - The regenerated dataset $D_S'$  is a stand-alone dataset.
> - It can be directly used by any CDR backbone.
> - Empirically, we observe consistent improvements across three diverse models: LightGCN, BiTGCF, and CUT.
>
>
> **Summary**
>
> Although [1–3] introduce valuable data-centric techniques, they focus on sample-level augmentation within model training. Our work instead performs dataset-level regeneration, enabled by an offline causal filtering mechanism and produces a reusable, model-agnostic dataset that improves a wide range of CDR models. This represents a fundamentally different problem setting, methodology, and capability compared with existing data-centric approaches.
>
>
> **Comment:**
> > There is an inconsistency between the paper title in the main text and the title on OpenReview. The authors should ensure that the titles are consistent before submission.
>
> **Ａ2:** Thank you for pointing this out. We revised our paper's title in the revision version.
>
> **References:**
>
> [1]Zhu, Jiajie, et al. "Domain disentanglement with interpolative data augmentation for dual-target cross-domain recommendation." Proceedings of the 17th ACM Conference on Recommender Systems. 2023.
>
> [2]Ning, Xuying, et al. "i $^ 2$ VAE: Interest Information Augmentation with Variational Regularizers for Cross-Domain Sequential Recommendation." arXiv preprint arXiv:2405.20710 (2024).
>
> [3]Mao, Qingyang, et al. "Cross-reconstructed augmentation for dual-target cross-domain recommendation." Proceedings of the 47th International ACM SIGIR Conference on Research and Development in Information Retrieval. 2024.

---

> ### Author Response · Authors · 2025-11-19
> **Clarification of the novelty**
>
> **Comment:**
>
>
> > For the generation stage, the authors employ self-supervised pretraining, which is a common practice in graph learning, and therefore this stage lacks significant novelty.
> For the filtering stage, the authors adopt counterfactual interaction filtering. It would be helpful to clarify the motivation for using this technique compared with existing filter-based methods. Are there unique challenges that the counterfactual approach specifically addresses?
>
> **Response:**
>
> We thank the reviewer for the helpful comments.
>
> **1. Novelty of Generation Stage**
>
>
> Although our framework uses a self-supervised pretraining module, its role is not generic graph pretraining. Its purpose is to **construct a cross-domain surrogate model** specifically optimized to
>
> (i) learn transferable patterns from overlapping users and
>
> (ii) enable reliable counterfactual evaluation in the second stage.
>
> Unlike standard GNN pretraining, our masking strategy is applied only to overlapping users (Sec. 2.2) and forces the model to learn domain-bridging representations rather than general node embeddings. This surrogate model is later frozen and directly serves as the causal mechanism $F_\theta^T$ in Eq. Pr for evaluating edge-level counterfactual effects. Without this tailored surrogate, the filtering stage cannot be performed.
>
> Thus, the contribution is not using existing pretraining strategy, but designing pretraining **specifically to support data regeneration and counterfactual filtering**, which is fundamentally different from existing SSL practices.
>
> **2. Motivation for counterfactual interaction filtering**
>
> Existing filter-based CDR methods rely on heuristic criteria—similarity, reliability scores, transferability—none of which capture whether an interaction causally improves target-domain performance. This is crucial because CDR models can be misled by interactions that are factually correct in the source domain but causally harmful to the target domain.）(Fig. 1(b)).
>
> Our counterfactual approach (Sec. 2.3) is uniquely suited to this. It is target-oriented, answering not only the question "if an interaction is true or consistent in the source", but also: "Does this interaction (original or generated) have a positive causal effect on target-domain performance?" By quantifying interactions whose removal degrades target performance via inverse optimization (Eq. Pr), it purges causally harmful correlations while retaining beneficial cross-domain knowledge.

---

> ### Author Response · Authors · 2025-11-26
> **Gentle Reminder**
>
> Dear Reviewer eJuQ,
>
> As the discussion period is approaching its end, we wanted to kindly check if our previous response clarified your questions. We are happy to provide any further details if needed.
>
> Thank you for your time and effort in reviewing our paper.
>
> Best regards, Authors

---

### Official Review · Reviewer_p3jn · 2025-11-02

**Soundness:** 3
**Presentation:** 2
**Contribution:** 3
**Rating:** 6
**Confidence:** 3

**Summary:**

This paper proposes a dataset enhancement strategy for cross-domain recommendation models. It aims to address the sparse cross-domain user overlap and noisy cross-domain signal (negative transfer) issues. The proposed strategy takes two stages. The first stage generates more user-item connections to address the sparse cross-domain user overlap issue, by learning a model that reconstructs edges in the source domain user-item graph. The second stage learns to identify spurious edges in the source domain user-item graph which should be removed to mitigate the negative transfer issue. Experimental results on two commonly used datasets, Amazon and Douban, showed the effectiveness of the proposed strategy.

**Strengths:**

S1. The paper is motivated well with a detailed example to illustrate issues of existing cross-domain recommendation solutions.

S2. The proposed technique works on the dataset level and is orthogonal to cross-domain recommendation models, which has the potential to be applied to and strengthen different cross-domain recommendation models.

S3. The proposed technique is shown to be effective on commonly used benchmark datasets.

S4. Source code is available.

**Weaknesses:**

W1. Technical details:

- The synthetic edge set contains edges between every non-overlapping user and their top-$k$ relevant items in the source set. Even the top-$k$ items might not be very relevant for some of the users, and hence there may be false positives. Using a fix $k$ for all users might not be the most effective. How about using a score threshold to filter the items instead (or a combination of both)? Also, how is the value of $k$ chosen in the experiments, and how does its value impact overall accuracy?

- The NP-hardness of Problem $\overline{P}$ needs a proof.

- How are the node embeddings in $\mathcal{F}_\theta^T$ initialized?

W2. Experiments:

- The performance gains obtained by using the proposed Gen/Del dataset preparation strategy is quite small as shown in Table 1 (noting the statistical significance test results). The second-best results in the two N columns of the Douban datasets didn't seem to be labeled correctly.

- It would be interesting to see model running time results, model effectiveness results as $K$ (as in Recall/NDCG@$K$) varies, and model effectiveness results as the number of cross-domain overlapping users varies.

W3. Presentation:

- The preliminaries section should be moved to the main text to set up the context for the methodology section. Without it, the methodology section is difficult to follow.

- Even with the preliminaries section, the paper needs a notation table to explain what the many symbols mean in the paper.

- The final sentence in Appendix A, "The next section details the optimization techniques used to implement this filtering, integrating the pre-trained prediction model with edge weight adjustments to achieve the desired causal pruning.", seems to be disconnected from the subsequent section.

- Typo: "”science fiction”" => "``science fiction”"; "in the Appendix B" => "in Appendix B"; "The single-domain baselines, trained exclusively on the target dataset" => "The single-domain baselines are trained exclusively on the target dataset"

**Questions:**

Please refer to the Weaknesses section.

---

> ### Author Response · Authors · 2025-11-19
> **Response to Reviewer's Concern about the Top-K Selection Strategy in Generation**
>
> **Question:**
>
>
> > The synthetic edge set contains edges between every non-overlapping user and their top-k relevant items in the source set. Even the top-items might not be very relevant for some of the users, and hence there may be false positives. Using a fix for all users might not be the most effective. How about using a score threshold to filter the items instead (or a combination of both)? Also, how is the value of K chosen in the experiments, and how does its value impact overall accuracy?
>
>
> **Answer:** Thank you for this insightful comment and it directly targets a core design choice in our generation module.
>
>
> You are correct that a fixed top-k can introduce false-positive synthetic edges for users whose predicted affinities to the source domain are inherently weak (low top scores). In isolation, the generation phase alone would indeed benefit from a score threshold (absolute or relative) or a hybrid (e.g., top-k only among items exceeding a minimum score τ) to reduce such noise upfront.
>
>
> However, we chose uniform top-k **because the subsequent counterfactual filtering stage is explicitly designed to prune not only original spurious interactions but also any low-quality/false-positive edges introduced during generation.** The generation phase is intentionally **“loose” to create a sufficiently large pool of candidate bridges that maximally alleviates sparse overlap** (ensuring every non-overlapping target user gains meaningful connectivity to the source domain), while **relying on the causal filtering to remove non-causal or irrelevant synthetic edges afterward**. This generate-and-filter paradigm allows the two stages to complement each other: generation solves the “quantity” problem (too few bridges), and filtering solves the “quality” problem (spurious or weakly relevant bridges).
>
>
> Using a strict score threshold in the generation phase would risk being overly conservative because some users would end up with very few or even zero synthetic edges,which will make them still isolated. Our filtering, by contrast, is adaptive per-edge, so it naturally discards low-relevance synthetic edges while retaining the truly causal ones, without requiring manual thresholding.
>
>
> We also think combination  of topK and threshold works well and we conducted additional experiments comparing three generation strategies on BiTGCF+Gen/Del: (1) pure Top-K(Our methods), (2) pure score threshold, and (3) hybrid (Top-K only among items above threshold). K is setted as 40 both here and that in our paper and threshold are tuning from {0.3,0.5,0.7}. The results on four Amazon cross-domain scenarios are shown below:
>
> | Dataset            | TopK      |          | Threshold |          | Hybrid    |          |
> |--------------------|-----------|----------|-----------|----------|-----------|----------|
> |                    | Recall@10 | NDCG@10  | Recall@10 | NDCG@10  | Recall@10 | NDCG@10  |
> | cloth → sports     | 0.0692    | 0.0378   | 0.0684    | 0.0374   | 0.0685    | 0.0379   |
> | sports → cloth     | 0.0532    | 0.0297   | 0.0527    | 0.0290   | 0.0538    | 0.0295   |
> | cloth → video      | 0.1389    | 0.0766   | 0.1380    | 0.0752   | 0.1381    | 0.0753   |
> | video → cloth      | 0.0498    | 0.0278   | 0.0480    | 0.0269   | 0.0485    | 0.0274   |
>
>
> We can find in most of the cases, TopK strategy behaves the best.

---

> ### Author Response · Authors · 2025-11-19
> **Proof of NP-hradness; Initialization strategy and response to Response to Reviewer's Concern about Table 1**
>
> **Comment:**
>
>
> > The NP-hardness of Problem $\bar{P}$  needs a proof.
>
> Thank you for this important comment. We agree that the original submission did not clearly justify the NP-hardness of our problem, and we have since devoted substantial effort to providing a rigorous proof (now included as Lemma 1 in Appendix D). Here we briefly summarize the core idea.
>
> We work with the decision version $(P)d$ of our problem and reduce from the decision version of Maximum Coverage, $(P_{\mathrm{cov}})d$, which is NP-complete. Given an instance $\langle \mathcal{G}, \mathcal{S}, k, B \rangle$ of $(P_{\mathrm{cov}})d$, we construct in polynomial time an instance of $(P)d$ as follows: each ground element $t \in \mathcal{G}$ becomes a target edge $e_t^T$, and each subset $S_\ell \in \mathcal{S}$ becomes a source edge $e_\ell \in \mathcal{E}^S$. We label all source edges as negative and set the budget $N_{\max} := k$, so the constraint becomes equivalent to selecting at most $k$ subsets.We then define a simple, explicitly computable $\mathcal{F}^T_\theta$ such that the objective of our problem satisfies
> $
> \mathcal{J}(\mathcal{E}^{S'}) = \Bigl|\bigcup_{e_\ell \in \mathcal{E}^{S'}} S_\ell\Bigr|,
> $
> i.e., it exactly equals the number of covered elements in the Maximum Coverage instance. Hence there exists a set E_S' with |E_S'(neg)| ≤ N_max and $\mathcal{J}(\mathcal{E}^{S'}) >=B
> $ if and only if the original Maximum Coverage instance is a YES-instance. This gives a polynomial-time many-one reduction $(P_{\mathrm{cov}})_d ≤p  (P)_d$, so $(P)_d$ is NP-complete. Consequently, the optimization problem $(P)$ (and its penalized form $(\bar{P}))$ is NP-hard.
>
>
> **Question**
>
> >How are the node embeddings $F_\theta^T$ initialized?
>
> **Answer:** We follow the initialization scheme of LightGCN. The only trainable parameters of $F_\theta^T$ are the layer-0 user and item embeddings, which are randomly initialized with Xavier initialization. We do not use any pre-trained embeddings or side information. Higher-layer embeddings are obtained solely through the (weighted) neighborhood aggregation described in our model, without additional transformation matrices or non-linear activations, in line with the “light” design of LightGCN.
>
>
> **Comment:**
> >The performance gains obtained by using the proposed Gen/Del dataset preparation strategy is quite small as shown in Table 1 (noting the statistical significance test results).
>
> **Answer:** Thank you for pointing this out. Our method is a model-agnostic dataset preparation strategy, so **its effect is best evaluated by comparing each backbone in its cross-domain version (C) with and without our Gen/Del modules**. To make this clearer, we provided an ablation in Appendix E.4 (Figure 3 and Table 4), where we explicitly report the relative gains of “+Our Method” over (C).
>
> For example, on the Amazon Sport→Cloth task, adding our method to CUT improves Recall/NDCG from 0.0441/0.0252 to 0.0546/0.0308, which corresponds to **+23.81% and +22.22%** over the cross-domain baseline. On the same task, BiTGCF improves from 0.0508/0.0286 to 0.0532/0.0297 (**+4.72% / +3.85%**), and LightGCN improves from 0.0421/0.0231 to 0.0458/0.0262 (**+8.79% / +13.42%**). On Amazon Video→Cloth, BiTGCF improves from 0.0411/0.0230 to 0.0498/0.0278 (**+21.17% / +20.87%**), and CUT from 0.0381/0.0213 to 0.0463/0.0256 (**+21.52% / +20.19%**). These results show that, relative to strong cross-domain backbones, our dataset regeneration brings consistent and sometimes substantial gains, especially on the more challenging asymmetric domain pairs.
>
> **Comment:**
> >The second-best results in the two N columns of the Douban datasets didn't seem to be labeled correctly.
>
> **Answer:**
> Thanks for pointed out. You are right that the underline formatting is incorrect in the current draft. We have corrested it in the Rebuttal Revision.

---

> ### Author Response · Authors · 2025-11-19
> **Running time results and results of Recall/NDCG@K**
>
> **Comment:**
> >It would be interesting to see model running time results, model effectiveness results as K (in Recall/NDCG@K) varies, and model effectiveness results as the number of cross-domain overlapping users varies.
>
>
> **Answer:** We thank the reviewer for this suggestion.
> As for model running time results, since the preprocessing pipeline of our method is identical across all backbone models, we report the runtime of our method when applied to different datasets (independent of the downstream CDR model).
> Note that:
>
> “Generation – Training” and “Filtering – Training” refer to the time taken to train the generation model and the filtering model for one epoch, respectively.
> “Generation – Inference” and “Filtering – Inference” refer to the actual data-processing (forward-pass) time for constructing the final synthetic edges.
>
> As the measured wall-clock time can vary slightly across runs and hardware, we additionally provide a detailed theoretical analysis of the time and space complexity of our method in Appendix F.
>
>
> | Dataset              | Cloth→Sports | Sports→Cloth | Cloth→Video | Video→Cloth | Movie↔Music |
> |----------------------|--------------|--------------|-------------|-------------|-------------|
> | Generation – Training| 0.52 s       | 0.51 s       | 0.14 s      | 0.12 s      | 10.46 s     |
> | Generation – Inference| 0.21 s       | 0.18 s       | 0.18 s      | 0.16 s      | 0.27 s      |
> | Filtering – Training | 2.85 s       | 1.79 s       | 1.92 s      | 1.59 s      | 29.47 s     |
> | Filtering – Inference| 0.009 s      | 0.009 s      | 0.006 s     | 0.008 s     | 0.04 s      |
>
> As for the results of Recall@K and NDCG@K, we additionally evaluated K ∈ {5, 10, 20, 50}. The complete results are presented on the following two pages.

---

> ### Author Response · Authors · 2025-11-19
> **Results of Recall/NDCG@K**
>
> | Dataset | Domain pair   | Metric     | CUT    | CUT+gen/del | BiTGCF | BiTGCF+gen/del | LightGCN | LightGCN+gen/del |
> |---------|---------------|------------|--------|-------------|--------|----------------|----------|------------------|
> | Amazon  | Cloth-Sport   | Recall@5   | 0.0438 | 0.0448      | 0.0423 | 0.0438         | 0.0389   | 0.0417           |
> |         |               | Recall@10  | 0.0653 | 0.0652      | 0.0655 | 0.0692         | 0.0604   | 0.0632           |
> |         |               | Recall@20  | 0.0921 | 0.0946      | 0.0983 | 0.1025         | 0.0923   | 0.0945           |
> |         |               | Recall@50  | 0.1418 | 0.1475      | 0.1609 | 0.1652         | 0.1505   | 0.1523           |
> |         |               | NDCG@5     | 0.0301 | 0.0306      | 0.0285 | 0.0295         | 0.0262   | 0.0284           |
> |         |               | NDCG@10    | 0.0364 | 0.0366      | 0.0360 | 0.0378         | 0.0331   | 0.0354           |
> |         |               | NDCG@20    | 0.0439 | 0.0448      | 0.0444 | 0.0463         | 0.0413   | 0.0434           |
> |         |               | NDCG@50    | 0.0538 | 0.0555      | 0.0569 | 0.0588         | 0.0530   | 0.0550           |
> |         | Sport-Cloth   | Recall@5   | 0.0298 | 0.0377      | 0.0349 | 0.0360         | 0.0257   | 0.0307           |
> |         |               | Recall@10  | 0.0441 | 0.0546      | 0.0508 | 0.0532         | 0.0385   | 0.0458           |
> |         |               | Recall@20  | 0.0623 | 0.0744      | 0.0749 | 0.0756         | 0.0565   | 0.0660           |
> |         |               | Recall@50  | 0.0988 | 0.1076      | 0.1184 | 0.1174         | 0.0882   | 0.1034           |
> |         |               | NDCG@5     | 0.0199 | 0.0253      | 0.0235 | 0.0241         | 0.0167   | 0.2130           |
> |         |               | NDCG@10    | 0.0252 | 0.0308      | 0.0286 | 0.0297         | 0.0207   | 0.0262           |
> |         |               | NDCG@20    | 0.0290 | 0.0358      | 0.0347 | 0.0353         | 0.0256   | 0.0313           |
> |         |               | NDCG@50    | 0.0363 | 0.0424      | 0.0434 | 0.0436         | 0.0318   | 0.0387           |
> |         | Cloth-Video   | Recall@5   | 0.0845 | 0.0877      | 0.0859 | 0.0882         | 0.0794   | 0.0863           |
> |         |               | Recall@10  | 0.1303 | 0.1325      | 0.1311 | 0.1389         | 0.1181   | 0.1344           |
> |         |               | Recall@20  | 0.1881 | 0.1913      | 0.1963 | 0.2013         | 0.1836   | 0.1929           |
> |         |               | Recall@50  | 0.2950 | 0.2949      | 0.3081 | 0.3157         | 0.2916   | 0.3037           |
> |         |               | NDCG@5     | 0.0577 | 0.0593      | 0.0568 | 0.0590         | 0.0518   | 0.0576           |
> |         |               | NDCG@10    | 0.0720 | 0.0740      | 0.0715 | 0.0766         | 0.0639   | 0.0733           |
> |         |               | NDCG@20    | 0.0875 | 0.0890      | 0.0882 | 0.0916         | 0.0826   | 0.0883           |
> |         |               | NDCG@50    | 0.1092 | 0.1101      | 0.1109 | 0.1148         | 0.1028   | 0.1108           |
> |         | Video-Cloth   | Recall@5   | 0.0248 | 0.0306      | 0.0296 | 0.0333         | 0.0253   | 0.0285           |
> |         |               | Recall@10  | 0.0381 | 0.0463      | 0.0411 | 0.0498         | 0.0385   | 0.0422           |
> |         |               | Recall@20  | 0.0582 | 0.0642      | 0.0636 | 0.0695         | 0.0557   | 0.0599           |
> |         |               | Recall@50  | 0.0942 | 0.1024      | 0.0968 | 0.1095         | 0.0893   | 0.0952           |
> |         |               | NDCG@5     | 0.0162 | 0.0206      | 0.0186 | 0.0225         | 0.0175   | 0.0192           |
> |         |               | NDCG@10    | 0.0213 | 0.0256      | 0.0230 | 0.0278         | 0.0207   | 0.0236           |
> |         |               | NDCG@20    | 0.0263 | 0.0302      | 0.2940 | 0.0328         | 0.0260   | 0.0281           |
> |         |               | NDCG@50    | 0.0336 | 0.0377      | 0.3580 | 0.0407         | 0.0327   | 0.0350           |

---

> ### Author Response · Authors · 2025-11-19
>
> | Dataset | Domain pair   | Metric     | CUT    | CUT+gen/del | BiTGCF | BiTGCF+gen/del | LightGCN | LightGCN+gen/del |
> |---------|---------------|------------|--------|-------------|--------|----------------|----------|------------------|
> | Douban  | Movie-Music   | Recall@5   | 0.0786 | 0.0809      | 0.0793 | 0.0802         | 0.0712   | 0.0728           |
> |         |               | Recall@10  | 0.1205 | 0.1248      | 0.1228 | 0.1237         | 0.1069   | 0.1097           |
> |         |               | Recall@20  | 0.1816 | 0.1842      | 0.1797 | 0.1826         | 0.1738   | 0.1764           |
> |         |               | Recall@50  | 0.2824 | 0.2856      | 0.2824 | 0.2825         | 0.2618   | 0.2654           |
> |         |               | NDCG@5     | 0.0824 | 0.0856      | 0.0751 | 0.0810         | 0.0612   | 0.0628           |
> |         |               | NDCG@10    | 0.0946 | 0.0962      | 0.0918 | 0.0926         | 0.0806   | 0.0785           |
> |         |               | NDCG@20    | 0.1124 | 0.1167      | 0.1023 | 0.1070         | 0.0889   | 0.8860           |
> |         |               | NDCG@50    | 0.1296 | 0.1326      | 0.1263 | 0.1293         | 0.1024   | 0.1026           |
> |         | Music-movie   | Recall@5   | 0.0896 | 0.0912      | 0.0823 | 0.0869         | 0.0731   | 0.0728           |
> |         |               | Recall@10  | 0.1393 | 0.1396      | 0.1289 | 0.1362         | 0.1031   | 0.1024           |
> |         |               | Recall@20  | 0.2078 | 0.2086      | 0.1930 | 0.2028         | 0.1612   | 0.1654           |
> |         |               | Recall@50  | 0.3364 | 0.3384      | 0.3139 | 0.3261         | 0.2882   | 0.2865           |
> |         |               | NDCG@5     | 0.1228 | 0.1248      | 0.1128 | 0.1168         | 0.1076   | 0.1062           |
> |         |               | NDCG@10    | 0.1437 | 0.1442      | 0.1219 | 0.1275         | 0.1096   | 0.1008           |
> |         |               | NDCG@20    | 0.1446 | 0.1468      | 0.1336 | 0.1402         | 0.1256   | 0.1248           |
> |         |               | NDCG@50    | 0.1658 | 0.1684      | 0.1582 | 0.1658         | 0.1468   | 0.1456           |

---

> ### Author Response · Authors · 2025-11-19
> **Response to the presentation problems**
>
> **Comment:**
> >The preliminaries section should be moved to the main text to set up the context for the methodology section. Without it, the methodology section is difficult to follow.
>
> **Respond:**
> Thanks for the suggestion. We moved the original Related Work section to Appendix A to avoid redundancy and moved the Preliminaries section before the Methodology section in the latest version.
>
> **Comment:**
> >Even with the preliminaries section, the paper needs a notation table to explain what the many symbols mean in the paper.
>
> **Respond:**
> Thanks for the suggestion. We have compiled a complete list of all symbols used throughout the paper in the Notations section of Appendix C.
>
>
> **Comment:**
> >The final sentence in Appendix A, "The next section details the optimization techniques used to implement this filtering, integrating the pre-trained prediction model with edge weight adjustments to achieve the desired causal pruning.", seems to be disconnected from the subsequent section. Typo: "”science fiction”" => "``science fiction”"; "in the Appendix B" => "in Appendix B"; "The single-domain baselines, trained exclusively on the target dataset" => "The single-domain baselines are trained exclusively on the target dataset"
>
> **Respond:**
> Thanks for pointing out the typos. We followed the comments and revised them in the latest version.

---

> ### Author Response · Authors · 2025-11-26
> **Gentle Reminder**
>
> Dear Reviewer p3jn,
>
> As the discussion period is approaching its end, we wanted to kindly check if our previous response clarified your questions. We are happy to provide any further details if needed.
>
> Thank you for your time and effort in reviewing our paper.
>
> Best regards, Authors

---

> > ### Comment · Reviewer_p3jn · 2025-11-26
> >
> > Thank you for the detailed response. My questions are addressed. I'm raising my overall rating from 6 to 8.

---

> > > ### Author Response · Authors · 2025-11-26
> > >
> > > Dear reviewer p3jn,
> > >
> > > We appreciate your engagement and positive feedback. Thank you for acknowledging our rebuttal and raising the score. Your comments have helped us significantly improve the quality of our manuscript.

---

### Comment · Area_Chair_SBHf · 2025-11-27

Dear Authors and Reviewers,

The discussion phase will end soon. If you want to further discuss comments and replies with each other, please post your thoughts by adding official comments.

Thanks for your efforts and contributions to ICLR 2026.

Best regards,

Your Area Chair

---

### Meta-Review · Area_Chair_GXEJ · 2026-01-09

**Summary:**

A key problem in the recommendation domain is the lack of a standard data processing protocol. Another principle in the recommendation domain is keeping the data real with manual operations as little as possible so that the insight and conclusion on a dataset can benefit online performance. The idea of regenerating recommendation is thus quite risky, which may mislead the development of a direction. It is thus worthwhile to be much more carefully validated than the content in this manuscript.

**Reviewer Concerns:**

Some concerns.

**Reviewer Scores:**

Unlike.

---

### Decision · Program_Chairs · 2026-01-26

Reject